# Non-stem cell lineages as an alternative origin of intestinal tumorigenesis in the context of inflammation

Mathijs P. Verhagen [1], Rosalie Joosten[1], Mark Schmitt[1,2], Niko Välimäki [3,4], Andrea Sacchetti[1], Kristiina Rajamäki [3,4], Jiahn Choi [5], Paola Procopio[2], Sara Silva[2], Berdine van der Steen[6], Thierry P. P. van den Bosch[1], Danielle Seinstra[1], Annemarie C. de Vries[7], Michail Doukas [1], Leonard H. Augenlicht[5], Lauri A. Aaltonen [3,8] & Riccardo Fodde [1] ✉

According to conventional views, colon cancer originates from stem cells. However, inflammation, a key risk factor for colon cancer, has been shown to suppress intestinal stemness. Here, we used Paneth cells as a model to assess the capacity of differentiated lineages to trigger tumorigenesis in the context of inflammation in mice. Upon inflammation, Paneth cell-specific *Apc* mutations led to intestinal tumors reminiscent not only of those arising in patients with inflammatory bowel disease, but also of a larger fraction of human sporadic colon cancers. The latter is possibly because of the inflammatory consequences of western-style dietary habits, a major colon cancer risk factor. Machine learning methods designed to predict the cell-of-origin of cancer from patient-derived tumor samples confirmed that, in a substantial fraction of sporadic cases, the origins of colon cancer reside in secretory lineages and not in stem cells.

The origin of the vast majority of cancers is thought to reside in stem-like or progenitor-like cells, which satisfy the need for active proliferation, self-renewal and differentiation capacity[1–3]. This was demonstrated in the mouse intestine where loss-of-function mutations in the *Apc* tumor suppressor gene successfully initiate adenoma formation only when they occur in *Lgr5*⁺ intestinal stem cells (ISCs). When the same *Apc* mutation is introduced in more-committed and shorter-lived transit-amplifying cells, tumor formation is halted at early microadenoma stages[2]. However, in addition to this 'bottom-up' scenario, 'top-down' models of intestinal tumorigenesis have been proposed in which more-committed intestinal cells located at higher positions along the crypt–villus axis are likely to initiate cancer, especially

in the context of tissue injury and inflammation[4,5]. In the specific case of human colon cancer, western-style dietary habits and chronic inflammation, two of the major etiologic factors associated with increased risk for sporadic malignant disease in the digestive tract, are thought to induce specific cellular and molecular alterations in the intestinal epithelium that ultimately lead to the expansion of cell targets for tumor initiation and progression[4,6]. However, which specific cell lineages are capable of dedifferentiating upon tissue injury and which are the underlying mechanisms remain largely unclear.

Paneth cells (PCs) are specialized secretory cells located at the very bottom of the crypt of Lieberkühn in the small intestine where they secrete antimicrobial peptides into the lumen[7]. Moreover, they provide

[1]Department of Pathology, Erasmus University Medical Center, Rotterdam, The Netherlands. [2]Institute of Pharmacology, University of Marburg, Marburg, Germany. [3]Department of Medical and Clinical Genetics, University of Helsinki, Helsinki, Finland. [4]Applied Tumor Genomics Research Program, Research Programs Unit, University of Helsinki, Helsinki, Finland. [5]Department of Cell Biology, Albert Einstein College of Medicine, New York, NY, USA. [6]Department of Otorhinolaryngology and Head & Neck Surgery, Erasmus University Medical Center, Rotterdam, The Netherlands. [7]Department of Gastroenterology and Hepatology, Erasmus University Medical Center, Rotterdam, The Netherlands. [8]iCAN Digital Precision Cancer Medicine Flagship, University of Helsinki, Helsinki, Finland. ✉e-mail: r.fodde@erasmusmc.nl

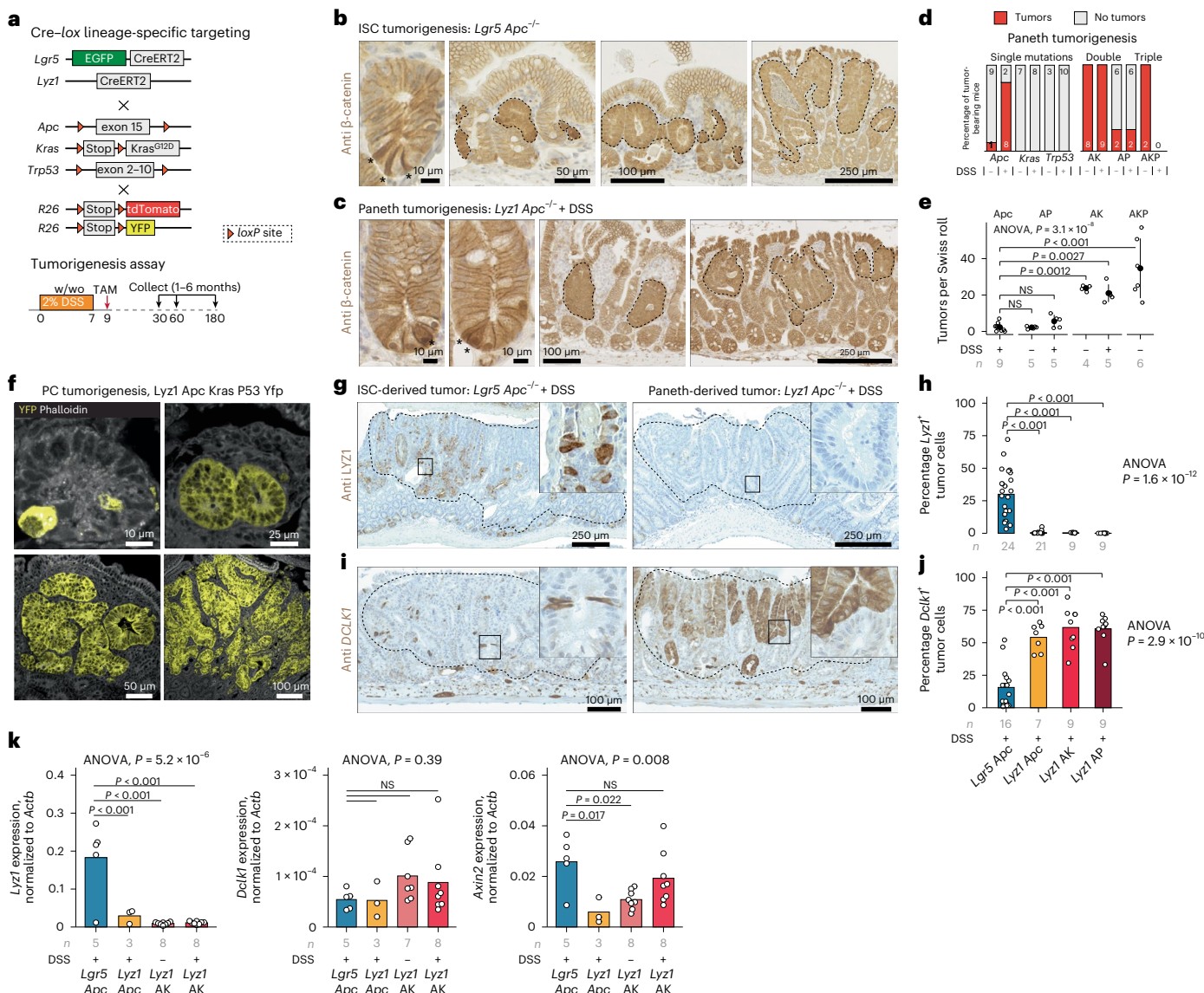

**Fig. 1 | PCs as the cell-of-origin of intestinal tumors in the context of inflammation and loss of *Apc*. a**, Cre–*lox* strategy aimed at the targeting of *Apc*, *Kras* and *Trp53* mutations in ISCs (*Lgr5*+ ISCs) and PCs (*Lyz1*+ PCs). w/wo indicates the presence (with) or absence (without) of DSS. **b,c**, β-Catenin IHC analysis of intestinal tumors initiated from *Lgr5*+ ISCs (**b**) and PCs (**c**). Asterisks indicate *Lgr5*+ ISCs and PCs with enhanced cytoplasmic and nuclear β-catenin accumulation; tumor foci and adenomas are indicated by dashed lines. **d,e**, Tumor multiplicity was calculated according to tumor-bearing mice (**d**) and by tumor number per genotype (**e**) in the presence/absence of DSS and based on Swiss roll counts.

Error bars denote s.d. *P* values denote one-way ANOVA and Tukey's post hoc tests for group comparisons. **f**, Lineage-tracing analysis of PCs, labeled using yellow fluorescent protein (YFP), at different stages of tumor initiation and progression. **g–j**, LYZ1 (**g**) and DCLK1 (**i**) IHC analysis of *Lgr5*+ ISC-derived (left) and PC-derived (right) adenomas, and quantification of number of *Lyz1*+ (**h**) and *Dclk1*+ (**j**) tumor cells. *P* values depict one-way ANOVA and Tukey's post hoc tests for group comparisons. **k**, *Lyz1*, *Dclk1* and *Axin2* quantitative PCR expression analysis across different adenoma genotypes. *P* values represent one-way ANOVA and Tukey's post hoc tests for group comparisons.

## Results

### PCs as the origin of intestinal tumors in the context of inflammation

We first reported on the ability of PCs to re-enter the cell cycle and dedifferentiate upon irradiation and inflammation to acquire stem cell-like features and contribute to the tissue regenerative response[11–13].

essential physical support and secrete signals to ensure *Lgr5*+ stem cell function[8]. In the colon, where PCs are not present, *Kit*+ Paneth-like cells, also known as deep crypt secretory cells (DCSs), have secretory and niche-like functional roles analogous to those of bona fide PCs in the small intestine[9,10]. Here, we used PCs as a model of a fully differentiated and postmitotic intestinal lineage capable of dedifferentiation and tumor formation in the context of inflammation.

Consequently, we questioned whether PCs could be the origin of intestinal cancer in the context of inflammation. We bred mice carrying *lox* alleles at the tumor suppressors and oncogenes most frequently mutated along the adenoma-to-carcinoma sequence, namely *Apc*[14], *Kras*[15] and *Trp53* (encoding p53)[16], each combined with Cre specific for *Lgr5*+ ISCs (*Lgr5*CreERT2-EGFP)[17] or for PCs (*Lyz1*CreERT2)[18]. Following Cre activation by tamoxifen, dextran sulfate sodium (DSS) was administered through drinking water to model inflammation (Fig. 1a). In the absence of DSS-induced inflammation, PC-specific single gene mutations did not give rise to intestinal tumors. By contrast, loss of *Apc* in *Lgr5*+ ISCs transformed crypts into β-catenin^hi foci that grew into multiple adenomas 4–6 weeks after Cre induction (Fig. 1b). When single gene mutations were combined with DSS administration, *Apc* loss in PCs resulted in increased nuclear and cytoplasmic β-catenin

expression eventually leading to the formation of PC-derived adenomas (Fig. 1c). Of note, Paneth-specific *Kras* or *Trp53* mutations did not result in tumor formation even in the presence of the inflammatory stimulus (Fig. 1d). However, the compound loss of *Apc* and oncogenic activation of *Kras* in PCs resulted in a striking increase in tumor multiplicity (6.1-fold) even in the absence of DSS (6.9-fold) (Fig. 1e). The combination of *Apc* and *Trp53* mutations in PCs also led to an increase in tumor multiplicity upon DSS administration (1.6-fold), although to a lesser extent when compared with the compound *Apc/Kras*-mutant genotype, possibly indicating a distinct mechanism underlying tumor onset in these mice. Indeed, phospho-histone H2A.X (Ser139) immuno-histochemistry (IHC) analysis confirmed an increase in DNA damage and chromosomal instability in *Trp53*-mutant tumors (Extended Data Fig. 1a). Targeting all three genes in PCs resulted in a very aggressive phenotype with high tumor multiplicity (10.1-fold) in the absence of the inflammatory stimulus (Fig. 1e). When compared with *Apc*-driven tumors that originated in PCs, the histology of adenomas from mice in which two or three genes were targeted revealed a progressive increase in dysplasia and invasive morphology (Extended Data Fig. 1b). The distribution of adenomas along the small intestine also followed distinct patterns with a prevalence of duodenal tumors in compound *Apc/Kras* tumors regardless of DSS (Extended Data Fig. 2a).

To validate the PC origin of the observed intestinal tumors, we bred *Lyz1*[CreERT2] mice with R26[LSL-tdTomato] or R26[LSL-YFP] reporters and traced their lineage upon tamoxifen-driven targeting of the *Apc, Kras* and *Trp53* mutations. As shown in Fig. 1f and Extended Data Fig. 2b, this confirmed the PC origin of the corresponding tumors by capturing the process from microscopic lesions to adenoma formation.

Overall, these results demonstrate that PCs can initiate intestinal adenomas upon genetic ablation of *Apc* in the context of inflammation. In combination with *Apc* loss, activation of oncogenic *Kras* or loss of *Trp53* function rescues the need for an inflammatory stimulus and results in increased PC-derived tumor multiplicities and progression to a malignant phenotype.

Next, we characterized lineage-specific markers in PC- and ISC-originated tumors using IHC. Of note, although cells expressing the PC marker lysozyme (LYZ1) were notable in *Lgr5*-derived tumors (*Lgr5/Apc*: 30.0% ± 18.5% positive tumor cells), they were almost absent in adenomas that originated from PCs (*Lyz1/Apc*: 0.48% ± 1.16%) (Fig. 1g,h). The opposite was observed for DCLK1 (doublecortin-like kinase 1), a Tuft[19] and tumor stem cell marker[20,21], that was more frequently detected among PC-derived adenomas (*Lyz1/Apc*: 54.1% ± 10.5%) when compared with *Lgr5*-derived tumors (*Lgr5/Apc*: 15.6% ± 15.7%) (Fig. 1i,j). Other lineage-specific markers for enteroendocrine (CHGcA), goblet (MUC2) and stem cells (OLFM4) showed variable levels without clear-cut differences among tumors with different cells-of-origin (Extended Data Fig. 2c). The increased *Dclk1* expression in PC-derived tumors is of interest in view of its association with increased immune and stromal infiltration in colon cancer[22].

To confirm these results at the transcriptional level, expression levels of *Lyz1* and *Dclk1* genes were analyzed by quantitative PCR with reverse transcription (Fig. 1k). Indeed, *Lyz1* expression was lower in Paneth-derived tumors (*Lgr5/Apc* versus *Lyz1/Apc*: $\log_2$-transformed fold change = 2.64, $P = 7.5 \times 10^{-4}$) when compared with *Lgr5*-derived tumors. *Dclk1* expression was very low and variable at the RNA level, and did not show significant differences across the groups.

To assess the relative activation of the WNT signaling pathway among the different tumor groups, we measured expression levels of *Axin2*, a well-established WNT downstream target. *Axin2* expression was higher in *Lgr5*-derived tumors compared with PC-derived tumors (*Lgr5/Apc* versus *Lyz1/Apc*: $\log_2$-transformed fold change = 2.12, $P = 0.017$) (Fig. 1k). Moreover, both *Kras* oncogenic activation and inflammation gradually increased *Axin2* levels in PC-derived tumors, in agreement with the previously reported synergism between *Apc* and *Kras* mutations in activation of the WNT pathway[23].

Thus, upon tumorigenesis, PCs dedifferentiate to a state that hampers secretory differentiation leading to specific patterns of tumor histology and gene expression distinct from that of *Lgr5*-derived tumors.

## PCs dedifferentiate into revival stem cells upon enhanced WNT signaling activation

To elucidate the mechanisms that underlie the conversion of PCs into cells-of-origin of small intestinal tumors in the context of inflammation and/or of specific genetic hits, we combined single-cell RNA sequencing (scRNA-seq) analysis with lineage tracing. To this end, we induced the *Apc, Kras* and *Trp53* genetic mutations in R26[LSL-tdTomato]/*Lyz1*[CreERT2] (or R26[LSL-YFP]) reporter strains in the presence or absence of DSS (Fig. 2a). Subsequently, cells were harvested from the intestinal epithelium, purified by FACS and transcriptionally profiled by scRNA-seq (Methods and Extended Data Fig. 3). After preprocessing, we obtained the transcriptomes of 23.231 epithelial cells from 32 mice, distributed over the different lineages of the intestinal epithelium (Fig. 2b). Close examination of cells positive for the reporter genes revealed novel clusters of PCs that arise upon DSS administration and specific gene mutations, but were not observed among PCs under homeostatic conditions (PC cluster 1–4, Fig. 2c).

To characterize the novel PC-derived states, we performed differential expression analysis and identified cluster-specific markers (Fig. 2d and Supplementary Table 1). Whereas PC cluster 1 appeared at low frequency across different genotypes, PC cluster 2 arises directly upon exposure to the inflammatory stimulus. Both PC cluster 1 and cluster 2 are characterized by increased expression of two markers of radio-resistant and secretory progenitors with self-renewal capacity during regeneration, namely *Krt19* (ref. 24) and *Atoh1* (ref. 25), whereas increased expression of *Reg3b*, known for its protective role in the development of colitis and ileitis[26], and *Cdkn1a* (encoding p21), a marker of terminal differentiation in the intestine[27], was observed in cluster 2 compared with cluster 1.

PC cluster 3 became apparent in mice carrying *Apc* mutations (7.23% ± 5.77% of traced cells) alone and in combination with DSS treatment (16.40% ± 2.10% of traced cells), and in double- and triple-mutant (AP, AK and AKP) mice, although not in mice carrying single *Kras* or *Trp53* mutations. PC cluster 3 showed increased expression of *Gif* (gastric intrinsic factor), *Cd81*, a tetraspanin family member known to mark the response to gamma-irradiation and correlated with the expression of ISC- and proliferation genes[28], and *Prom1* (also known as *CD133*), a well-established colon cancer stem cell marker[29].

PC cluster 4 consisted of cells from mice in which double (AK) and triple (AKP) mutations were targeted to PCs (23.25% ± 11.23% and 52.26% ± 28.23% of traced cells, respectively). Increased expression of *Anxa2* (encoding annexin 2), a functional marker of inflammatory response, and *Clu* (clusterin), previously shown to earmark revival stem cells (RSCs) upon gamma-irradiation[30], feature in PC cluster 4. Accordingly, evaluation of the RSC signature showed elevated expression among the PC clusters (Fig. 2e), and in situ hybridization analysis in PC-derived tumors from mice carrying compound mutations (AK and AKP) (Fig. 2f and Extended Data Fig. 4a,b) confirmed increased *Clu* expression. Finally, pathway analysis revealed the similarities between the PC-derived cluster 4 and RSCs, both earmarked by the activation of Yap1 signaling and specific inflammatory pathways (Fig. 2g). Compared with RSCs, PC-derived and *Apc/Kras*-mutant cells from cluster 4 showed increased levels of TGFβ and WNT signaling (Extended Data Fig. 4c).

Thus, upon genetic targeting or inflammatory stimulus, PCs escape their homeostatic identity and acquire distinct cellular features, as shown by scRNA-seq and FACS analysis (Extended Data Fig. 4d,e). Of note, DSS treatment led to lower expression of *Lgr5* and *Ascl2* in stem cells, as well as a lower association of the ISC signature, confirming our and others' previous observations that resident stem cells lose their multipotency upon acute inflammation[13] (Extended Data Fig. 4f).

Collectively, these findings demonstrate that PCs efficiently dedifferentiate upon genetic targeting or inflammatory stimulus

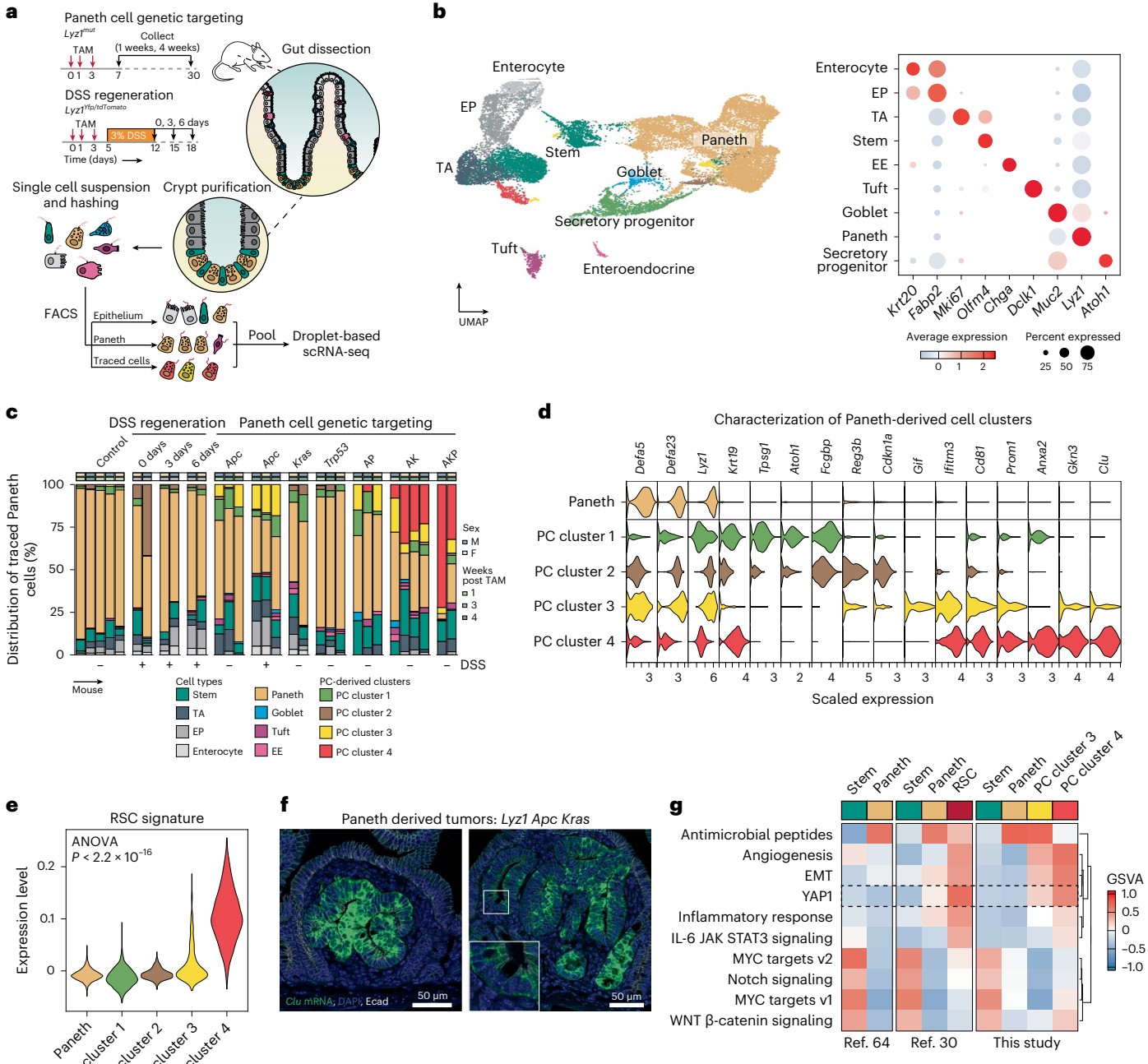

**Fig. 2 | PCs may dedifferentiate into a revival stem cell identity. a**, Schematics of the experimental approach, adapted from ref. 63, Springer Nature Limited. After genetic targeting of PCs, intestinal crypts were extracted, and the isolated cells were labeled with hashing antibodies and sorted according to three different strategies: epithelium, PC-enriched and PC-traced cells. **b**, UMAP embedding of the different cell clusters or lineages (left), annotated according to the expression of canonical marker genes (right). EE, enteroendocrine cells; EP, enterocytes progenitors; TA, transit-amplifying cells. **c**, Bar plot of the distribution of traced cells across the different mouse genotypes and experimental conditions. **d**, Violin plots representing marker genes of the newly identified Paneth-derived cell clusters (PC cluster 1–4). **e**, Association analysis of the RSC signature with PC cluster 1–4. The *P* value denotes the result of one-way ANOVA. **f**, RNA in situ hybridizations of the *Clu* gene in small tumors derived from PCs upon compound targeting of *Apc* and *Kras* mutations. **g**, Gene sets variation analysis among refs. 30,64 and the current study. EMT, epithelila-to-mesenchymal transition.

leading to distinct cellular identities. During tumorigenesis driven by *Apc/Kras*, PCs share features with the YAP1-dependent RSC identity, and further activate TGFβ and WNT signaling in their conversion to bona fide tumor cells.

**Transcriptomic comparison of Paneth- and *Lgr5*-derived tumors reveals a dichotomy in stem cell phenotypes**

To investigate the consequences of cell-of-origin identity on the transcriptional profile of the resulting intestinal tumors, we performed

bulk RNA sequencing (RNA-seq) of macroscopically dissected lesions originating from ISCs and PCs (Fig. 3a). Principal component analysis (PCA) revealed that the major variance component (61%) was attributed to differences in the cell-of-origin, whereas the impact of genotype or inflammatory stimulus became notable in the second component of variation (10%) (Fig. 3b). Differential expression analysis between tumors derived from PCs and ISCs in the same genetic and inflammatory context (*Apc* and DSS) revealed tumor signatures specific for each cell-of-origin (Supplementary Table 2).

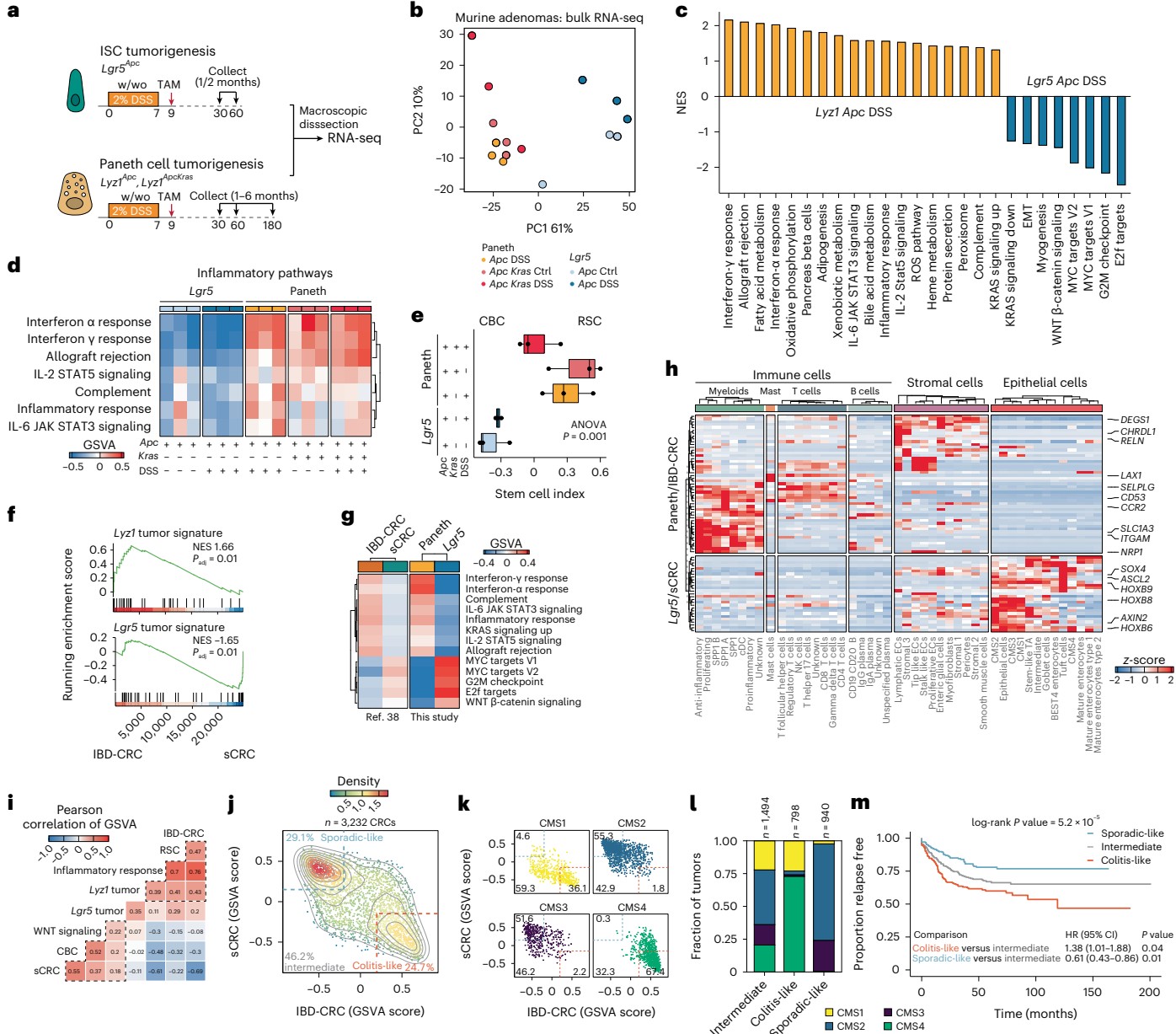

**Fig. 3 | Paneth-derived adenomas have an inflammatory phenotype mimicking colitis-associated colon cancer. a**, Schematics of the experimental approach to compare PC- and *Lgr5*-derived adenomas. **b**, PCA plot showing that the cell-of-origin is the dominant discriminator of variance. Ctrl, control. **c**, Bar plot summarizing the GSEA between Paneth-derived (*Lyz1 Apc* DSS) and ISC-derived (*Lgr5 Apc* DSS) tumors. Pathways were filtered based on *P* < 0.05 and absolute normalized enrichment score (NES) > 0.5. Pathways that are significant in the IBD-CRC versus sCRC comparison have bold labels. ROS, reactive oxygen species. **d**, Subset of inflammatory pathways, visualized as a heatmap based on values from the GSVA. **e**, Box plots showing results of the stem cell index. The *P* value depicts the result of one-way ANOVA. *n* = 3 biologically independent samples per group. Box plots display the median, lower and upper hinges corresponding to the first and third quartiles. Whiskers extend from the hinges to maximum or minimum values, no further than 1.5× interquartile range. **f**, GSEA showing a significant but opposite association between the *Lyz1* tumor signature and IBD-CRCs, and between the *Lgr5* tumor signature and sCRC. The *P* value denotes the Benjamini–Hochberg adjusted value of the two-sided enrichment *P* value from GSEA. **g**, Heatmap showing GSVA scores, averaged per tumor group,

of pathways with similar patterns between the mouse and human tumor groups. **h**, Heatmap highlighting differentially expressed genes (log₂-transformed fold change > 1.5, *P*adj < 0.01). Two-tailed *P* values were derived from the Wald test with Benjamini–Hochberg correction for multiple testing shared between the Paneth or IBD tumors and the *Lgr5* or sCRC tumors. Values denote *z*-scores of average expression per cell type. cDC, conventional dendritic cells; EC, endothelial cells; NK, natural killer cells; Ig, immunoglobulin. **i–m**, Two distinct sporadic colon cancer identities become apparent upon analysis of a large cohort of CRC tumors (*n* = 3,232 samples). **i**, Heatmap showing Pearson correlations of the GSVA scores. **j–k**, Scatter plot showing two distinct clusters, sporadic-like and colitis-like, in all colon cancers (**j**) and stratified according to their CMS (**k**). Gray lines indicate contours lines, and dashed lines show thresholds to classify tumors in colitis-like, sporadic-like and intermediate groups. **l**, Stacked bar plot analysis showing the distribution of CMS1 to CMS4 across colitis-like and sporadic-like colon cancers. **m**, Kaplan–Meier survival analysis for relapse-free survival. *P* values denote the result of the log-rank test and Cox regression models for univariate analyses. Hazard ratios (HR) and confidence intervals (CI) are shown for pairwise comparisons.

Gene set enrichment analysis (GSEA) indicated that tumors derived from ISCs were characterized by high levels of MYC and WNT signaling, whereas PC-derived adenomas showed higher levels of inflammatory pathways indicative of infiltration from the tumor microenvironment (TME) (Fig. 3c and Extended Data Fig. 5). Of note, the inflammatory characteristics of PC-derived tumors were observed also in mice in which *Apc* and *Kras* mutations were targeted to PCs in the absence of DSS-driven inflammation (Fig. 3d), indicating that specific mutant genotypes and type of cell-of-origin can trigger tumor initiation by mimicking the inflammatory context otherwise brought about by DSS.

Next, we used the ISC index[31] (Methods) to predict the relative proportions of RSC and crypt-base columnar (CBCs) stem cells in the intestinal tumors. In agreement with the scRNA-seq analysis, PC-derived tumors were RSC-enriched, whereas ISC-derived tumors consisted mainly of CBCs (Fig. 3e). Notably, the highest RSC contribution was observed in tumors originating from *Apc/Kras*-mutant PCs in the absence of inflammation, when compared with the equivalent genotype upon DSS administration. The latter is of relevance to dissect the relative contribution of inflammatory insult and somatic mutations in the dedifferentiation process leading to tumor initiation. As mentioned before, one of the primary effects of various forms of tissue injury to the intestinal epithelium is the loss of resident stem cells. In a study by Singh et al.[32], ablation of *Lgr5*+ ISCs was performed to study its consequences using scRNA-seq. Our analysis of these datasets revealed elevated expression of secretory genes in regenerated stem cells normally restricted to the PC lineage (Extended Data Fig. 6a). In parallel, upon *Lgr5*+ cell ablation, PCs and goblet cells partially activate the RSC program (Extended Data Fig. 6b), indicating that stem cell loss is sufficient to trigger plasticity and RSC reprogramming from PCs. Hence, to assess whether removal of the resident stem cells is sufficient to activate the lineage-tracing capacity of PCs in the absence of inflammatory injury, we implemented a similar diphtheria toxin receptor (DTR)-based *Lgr5*+ ablation experiment[33]. As shown in Extended Data Fig. 6c,d, lineage tracing from PCs was similar to that previously shown upon inflammation[13].

Together, these results indicate that the cell-of-origin embodies the major source of intertumor variability, and that the Paneth or ISC origin is reflected by the RSC- or CBC-like profile of the resulting tumors, respectively. Notably, inflammation-driven ISC loss activates dedifferentiation at the epithelial level.

## The transcriptional profile of Paneth-derived tumors mimics colitis-associated colorectal cancer

The small intestinal location of PCs and the tumors originating from them raises questions on the relevance of our study for colon cancer, one of the most frequent causes of morbidity and mortality because of malignancy. To explore the general applicability of our results, we first analyzed bulk RNA-seq and scRNA-seq data from two distinct studies[34,35] centered around the azoxymethane (AOM)/DSS mouse model of colitis-driven colon cancer[36]. This protocol relies on the oncogenic β-catenin mutations caused by AOM, which in combination with DSS-driven ulcerative colitis, result in multiple adenocarcinomas in the distal colon. As shown in Extended Data Fig. 6e (CIBERSORTx[37]; Methods), analysis of the bulk RNA-seq data[35] from AOM/DSS-derived colon tumors revealed an abundant subpopulation reminiscent of the RSC-like PC cluster 4. Moreover, analysis of scRNA-seq data[35] confirmed that AOM/DSS-derived colon tumors, when compared with their sporadic counterparts, were distinct in terms of the qualitative and quantitative composition of their TME, namely a pronounced presence of infiltrating immune cells and tumor-associated fibroblasts (Extended Data Fig. 6f). Accordingly, tumor cells derived in the context of AOM/DSS share transcriptional similarity with Paneth-derived tumor cells and RSCs (Extended Data Fig. 6g,h).

Hence, notwithstanding the small intestinal location of the PC-derived tumors, their gene expression signatures and overall inflammatory TME profiles are characteristic of colitis-associated carcinoma in the mouse.

Next, in view of the marked differences in the transcriptional profiles between mouse intestinal tumors with distinct cells-of-origin, we questioned whether similar differences distinguish human sporadic colon cancers from those that arise in the context of IBD. RNA-seq profiles of human microsatellite stable (MSS) sporadic colorectal cancers (sCRC, n = 38) and from patients with inflammatory bowel disease (IBD-CRC, n = 14) were interrogated[38]. GSEA of the most differentially expressed genes ($\log_2$-transformed fold change > 5, adjusted $P$ ($P_{adj}$) < 0.01) from the mouse tumors (*Lyz1* tumor signature, n = 27 genes; *Lgr5* tumor signature, n = 40 genes) revealed a significant association between the *Lyz* tumor signature and IBD-CRC (NES = 1.67, $P_{adj}$ = 6.2 × 10⁻³), whereas the *Lgr5*T profile was significantly associated with sCRC (NES = −1.62, $P_{adj}$ = 0.013) (Fig. 3f, Extended Data Fig. 7a–c and Supplementary Table 3). Evaluation of the hallmarks from the molecular signature database[39] revealed gene sets common to PC-derived tumors and IBD-related CRCs (interferon alpha/interferon gamma, inflammatory response, IL-6/IL-2 signaling, KRAS, complement, allograft rejection), and to *Lgr5*-derived tumors and sCRCs (MYC targets, G2M checkpoint, E2F targets and WNT β-catenin signaling) (Fig. 3g and Extended Data Fig. 7d).

We then compared the differentially expressed genes from PC-derived versus *Lgr5*-derived mouse tumors and human IBD-CRC versus sCRC (Paneth/IBD-CRC, n = 49 genes; *Lgr5*/sCRC, n = 27 genes) and visualized their expression across different cell types based on a large scRNA-seq CRC study[40] (Fig. 3h). Of note, the markers shared between PC-derived tumors and IBD-CRC were dominantly expressed in myofibroblasts (for example, *ITGAM*, *SLC1A3*), T cells (for example, *SELPLG*, *LAX1*) and stromal cells (for example, *CHRDL1*, *RELN*). By contrast, *Lgr5*-derived tumors/sCRC markers were mostly observed in epithelial cells (for example, *HOXB6*, *HOXB8*, *HOXB9*, *AXIN2*, *ASCL2*), indicating the difference in stromal composition among these tumors (Fig. 3h). Comparison of a set of gene signatures (Supplementary Table 4) in a large CRC cohort[41] confirmed the presence of two distinct identities (Fig. 3i,j): a colitis-like identity enriched with RSCs and prevalent in consensus molecular subtype 4 (CMS4) (67%) and CMS1 (36%) tumors; and a sporadic-like identity enriched with CBCs and common in CMS2 (55%) and CMS3 tumors (52%) (Fig. 3k,l and Extended Data Fig. 7e). Survival analysis revealed significant differences in relapse-free survival between the sporadic- and colitis-like CRC groups ($P$ = 5.2 × 10⁻⁵) (Fig. 3m). Thus, transcriptional signatures derived from small intestinal mouse tumors originating from PCs significantly overlap with those from human colon cancer that arose in the context of IBD, possibly revealing a common cell-of-origin in secretory lineages.

Next, we investigated whether the inflammatory profile observed in PC-derived tumors is earmarked by distinct immune cell populations. By means of gene set variation analysis (GSVA) with a tumor-infiltrating lymphocyte-specific gene signature[42], a statistically significant higher tumor-infiltrating lymphocytes score was observed in PC-derived versus *Lgr5*-derived mouse tumors (Extended Data Fig. 8a). When using deconvolution of our data with CIBERSORTx[37], a significantly higher score was observed for immune cells in PC-derived versus *Lgr5*-derived tumors (Extended Data Fig. 8b). Furthermore, when zooming in on defined immune cell types, a highly significant and unique enrichment of γδ⁺ T cells was observed in PC-derived tumors (Extended Data Fig. 8c,d), in contrast to the enrichment of regulatory T (T_reg) cells in *Lgr5*-derived tumors. These findings were in concordance with what we observed for colitis-like versus sporadic-like CRCs (Extended Data Fig. 8e,f). These observations were further validated by analysis of the scRNA-seq data obtained from mouse colonic tumors induced by AOM/DSS previously[35]. As shown in Extended Data Fig. 9f–h, a subpopulation of *Cd8*⁻/*Cd4*⁻ and *Pdcd1*⁺/*Il17a*⁺ T cells earmarks these tumors, which likely represent the counterpart of γδ⁺ T cells observed in colitis-like colon cancers.

## Western-style diet triggers an inflammatory response leading to dedifferentiation of secretory cells

The relative representation of patient-derived colon cancers whose expression profiles are reminiscent of the PC-derived mouse tumors (~25%; Fig. 3j) vastly exceeds the expected proportion of colon cancers arising in patients with an history of clinically manifest IBD (1–2%)[43]. One possible explanation for this apparent discrepancy may be that western-style dietary habits, often in combination with chronic over-nutrition and sedentary lifestyles, have been associated with a state of chronic metabolic inflammation, termed 'metaflammation'[44]. In particular, a link between the consumption of a diet high in fat and sugar and PC dysfunction has recently been demonstrated[45]. Moreover, a purified mouse diet that mimics western-style dietary habits and underlies the increased risk of colon cancer (NWD1)[46] was recently shown to induce a low degree of chronic intestinal inflammation and other mechanisms that define pathogenesis of human IBD[47]. Therefore, we hypothesized that etiological drivers of colon cancer other than IBD, including widespread western-style dietary habits, may underlie dedifferentiation and tumor onset mechanisms similar to those observed upon acute DSS-driven inflammation.

To provide support for this hypothesis, we first fed C57BL6/J mice for 3 months with the western-style (NWD1) and control (American Institute of Nutrition 76A (AIN-76A)) diets and compared the transcriptional response of PCs with that obtained upon DSS administration (Fig. 4a). We examined genes upregulated upon inflammatory stimuli (DSS signature; Methods), which showed variable but overall increased levels in NWD1-fed mice when compared with those on the control diet (Fig. 4b). Indeed, GSEA confirmed that the DSS signature was associated significantly with PCs exposed to NWD1 (NES = 2.99, $P_{adj}$ < 0.001), indicating that western-style dietary habits trigger an inflammatory-like response in PCs (Extended Data Fig. 9a). At the gene ontology level, the western-style diet activated signaling pathways related to the cell cycle (G2M checkpoint) and proliferation (mitotic spindle, MYC targets; Extended Data Fig. 9b), suggesting that PCs re-enter the cell cycle upon long-term exposure to a western-style diet, similar to what is observed in DSS-driven inflammation[13].

In view of the previously reported acquisition of stem-like features by PCs upon inflammation[13], we next used the organoid reconstitution assay (ORA)[12,48] to assess whether similar effects are exerted by the western-style diet. We first coincubated Paneth and $Lgr5^+$ cells from AIN-76A- and NWD1-fed mice in all four combinations. As shown in Fig. 4c, PCs from NWD1-fed mice significantly improved organoid formation independently of their reconstitution with $Lgr5^+$ cells from NWD1- or AIN-76A-fed mice, possibly indicative of a paracrine effect enhancing the well-established niche (ISC-supporting) role of PCs. However, single (that is, nonreconstituted with $Lgr5^+$ ISCs) PCs from NWD1-fed mice formed organoids more efficiently when compared with either PCs from AIN-76A-fed mice or with $Lgr5^+$ ISCs from both groups of mice. These ex vivo results were further validated by lineage-tracing analysis of PCs in R26^LSL-YFP^Lyz1^CreERT2 NWD1-fed mice that revealed extended yellow fluorescent protein (YFP)-labeled ribbons thus confirming their dedifferentiation and acquisition of stem-like features induced by the western-style dietary cues (Fig. 4d).

To focus in on the primary transcriptional response of PCs to NWD1, we took advantage of the scRNA-seq data generated previously[47] upon exposure to NWD1 (Fig. 4a). Within 4 days of switching mice to the NWD1 diet, a subset of PCs became apparent whose transcriptional profile was strongly associated with the DSS signature (labeled 'diet response cells' in Fig. 4e). Mirroring our previous observations obtained immediately upon DSS inflammatory stimulus, these WSD-responsive cells increased their transcriptomic diversity as measured by CytoTRACE[49] (Fig. 4f and Methods). After short exposure to NWD1, the diet-responsive cells acquired stem cell markers while retaining some secretory features (Fig. 4g). Comparative pathway analysis between the transcriptional response of PCs to DSS and NWD1 revealed similar upregulation of the WNT, MYC, Hedgehog and G2M checkpoint signaling pathways (Fig. 4h).

Because the observed western-style diet-driven changes in gene expression are likely exerted through epigenetic modifications, we next analyzed scATAC (single cell assay for transposase-accessible chromatin with high-throughput sequencing) data obtained in the framework described previously[47]. Similar to what was observed by scRNA-seq analysis, we identified a group of NWD1 diet-responsive PCs with significant epigenetic modifications, including a main cluster (52%) on mouse chromosome 11 (synthenic with human chromosomes 17 and 5) that encompasses a considerable fraction of the genes encoding for members of the WNT, PI3K–AKT and cell-cycle pathways (Fig. 4i–k and Extended Data Fig. 9c–f). The latter were previously shown by our laboratory to underlie PC dedifferentiation and the acquisition of stem-like features upon DSS-driven inflammation[13].

To assess the effects of the NWD1 diet on mouse colon, we searched for proliferating Paneth-like cells, also known as DCS cells[9,12]. Inflammation-driven cell-cycle activation in these allegedly postmitotic lineages was previously observed in small intestinal PCs of DSS-treated mice and of patients with Crohn's disease[13]. By co-staining colonic tissues with the secretory lineage marker wheatgerm agglutinin (WGA)[50] and Ki67, a significant increase in the number of proliferating secretory cells located at the crypt bottom was observed both in mice fed the NWD1 diet and, as a positive control, in those administered DSS in their drinking water (Fig. 5a–c).

## Secretory lineages as the cell-of-origin of human colon cancer

Collectively, our results reveal an alternative bottom-up route to intestinal tumorigenesis originating from PCs in the mouse small intestine and, allegedly, from secretory lineages in the human colon, triggered by inflammatory cues, as in IBD, or through western-style dietary factors. To validate the relevance of our mouse study in patient-derived colon cancer, we took advantage of new computational methods[51–53] developed to predict the cell-of-origin of tumors by matching the mutational density along the cancer genome with the profiles of epigenetic modifications characteristic of normal cell types[54] (Extended Data Fig. 10a). Although tumors from patients with a history of IBD have a similar genome-wide tumor mutational burden compared with their sporadic counterparts (Extended Data Fig. 10b), the presence of regions (genomic windows, Methods) that are differentially mutated between sCRC and IBD-CRC is suggestive of alternative mutational patterns (Extended Data Fig. 10c,d). Hence, we compared the individual mutational landscapes with the epithelial cell types of the colon to compute the putative cells-of-origin.

As shown in the Fig. 5a (COOBoostR[53], Methods), although the majority of the sporadic colon cancers appear to originate from stem cells (52%), among the IBD-related cases, goblet (40%) and BEST4 (28%) cells represent the prevalent cells-of-origin. Strikingly, a substantial proportion (>40%) of sporadic cases are also predicted to originate from non-stem lineages, namely goblet (22%), enterocytes (2%) and BEST4 cells (17%). The latter, named after the specific expression of the bestrophin 4 gene (*BEST4*), form a newly identified and as yet only partially characterized intestinal epithelial lineage with dual absorptive and secretory features[55–57]. IHC and scRNA-seq analysis in a cohort of patients with ulcerative colitis confirmed the presence of actively proliferating goblet and BEST4 cells (MUC2^+ and BEST4^+) (Fig. 5e,f), indicative of the primary effects of inflammation in these otherwise postmitotic cells, likely to precede their acquisition of stem-like features.

These results are indicative of the fraction of sporadic colon cancers whose expression profiles are reminiscent of the PC-derived tumors and IBD-CRCs (24.7%; Fig. 3j). Furthermore, the availability of RNA-seq data from a subset of the patient-derived tumors allowed us to confirm the prevalence of RSC-like expression profiles and the

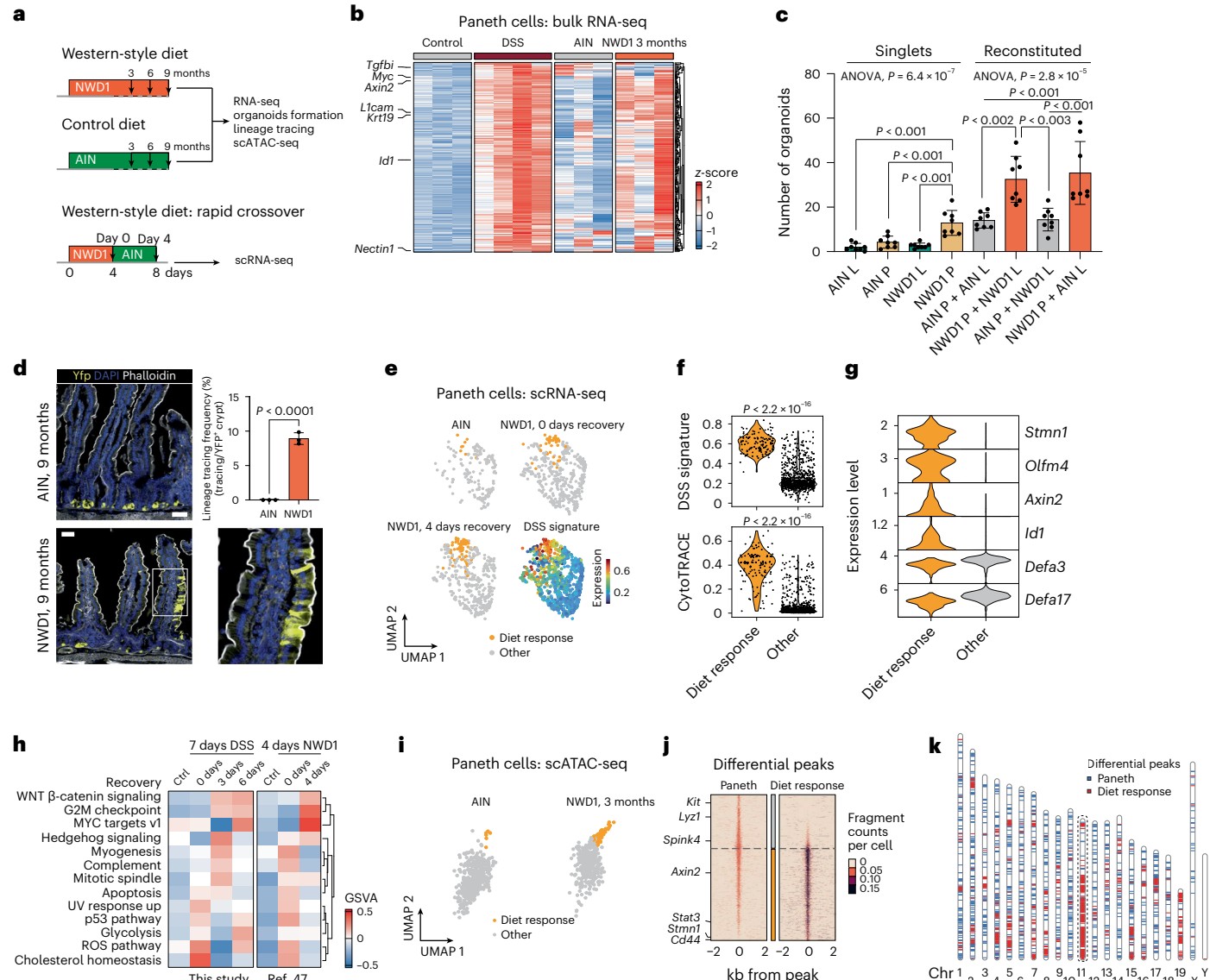

**Fig. 4 | Western-style diet triggers an inflammatory response leading to PC dedifferentiation. a**, Schematics of the experimental approach designed to investigate the consequences of short- and long-term exposure to a western-style diet (NWD1) versus control (AIN-76A) diets. **b**, Heatmap showing z-scored DSS signature (DSS versus control; $P_{adj}$ < 0.05, $\log_2$-transformed fold change > 0.25) in PCs exposed to DSS or NWD1. **c**, Organoid multiplicities derived either from single ISCs or PCs, and from reconstituted doublets (L, *Lgr5*⁺ ISCs; P, PCs). Pooled data from n = 4 independent experiments. P values were calculated using one-way ANOVA and Tukey's tests for group comparisons. Error bars depict s.d. **d**, Representative image of lineage tracings from a NWD1-fed *Lyz1*-YFP mouse. Scale bars, 50 μm. The P value depicts the result of a two-sided Student's t-test and the error bars represent s.d. Data from n = 3 mice. **e**, UMAP showing PCs from mice fed AIN-76A (AIN) and NWD1 (n = 3 mice per condition). The DSS signature

portrayed on UMAP embedding highlights a subcluster of PCs responsive to the NWD1 diet. **f**, Violin plots showing different levels of the DSS signature (top) and CytoTRACE score (bottom) between PCs responsive to the NWD1 diet and other PCs. P values show the significance of a two-sided Wilcoxon test. **g**, Violin plots representing marker genes of PCs responsive to the NWD1 diet, showing coexpression of stem and secretory markers. **h**, Heatmap visualization of GSVA, indicating pathways that are activated in PCs after exposure to DSS or NWD1. Comparison with data from ref. 47. **i**, UMAP plot of PC subset in the scATAC dataset of mice treated with AIN (n = 2) and NWD1 (n = 2). **j**, Heatmap listing differential peaks between diet response PCs and other PCs. **k**, Ideogram displaying the distribution along the mouse chromosomes (Chr) of the differential peaks observed upon diet response (red) when compared with those characteristic of PCs (blue).

high proportion of CMS1 and CMS4 among the sporadic and IBD cases predicted to originate from non-stem, and in particular goblet, cells (Fig. 5g,h and Extended Data Fig. 10e).

## Discussion

*Lgr5*⁺ stem cells have been established as the origin of intestinal tumors[2]. However, whether the same holds true in the context of inflammation has been challenged by their loss of lineage-tracing capacity and multipotency upon a broad spectrum of tissue injuries ranging from DSS-driven inflammation[13] to gamma-irradiation[11,24] and a western-style

diet[58]. Our results show that PC dedifferentiation and tumor onset in the mouse small intestine model sporadic colon cancer not only in patients with IBD, but also in a substantial proportion of cases predicted to originate in differentiated cell types and not in resident ISCs. Both the differential gene expression profiles and inflammatory and stromal features of the resulting tumors warrant a novel stratification of colon cancer for improved clinical management. Preliminary investigations (data not shown) revealed that candidate markers differentially expressed between PC- and *Lgr5*⁺-initiated mouse tumors such as DCLK1 and HOXB9 were not equally discriminative among

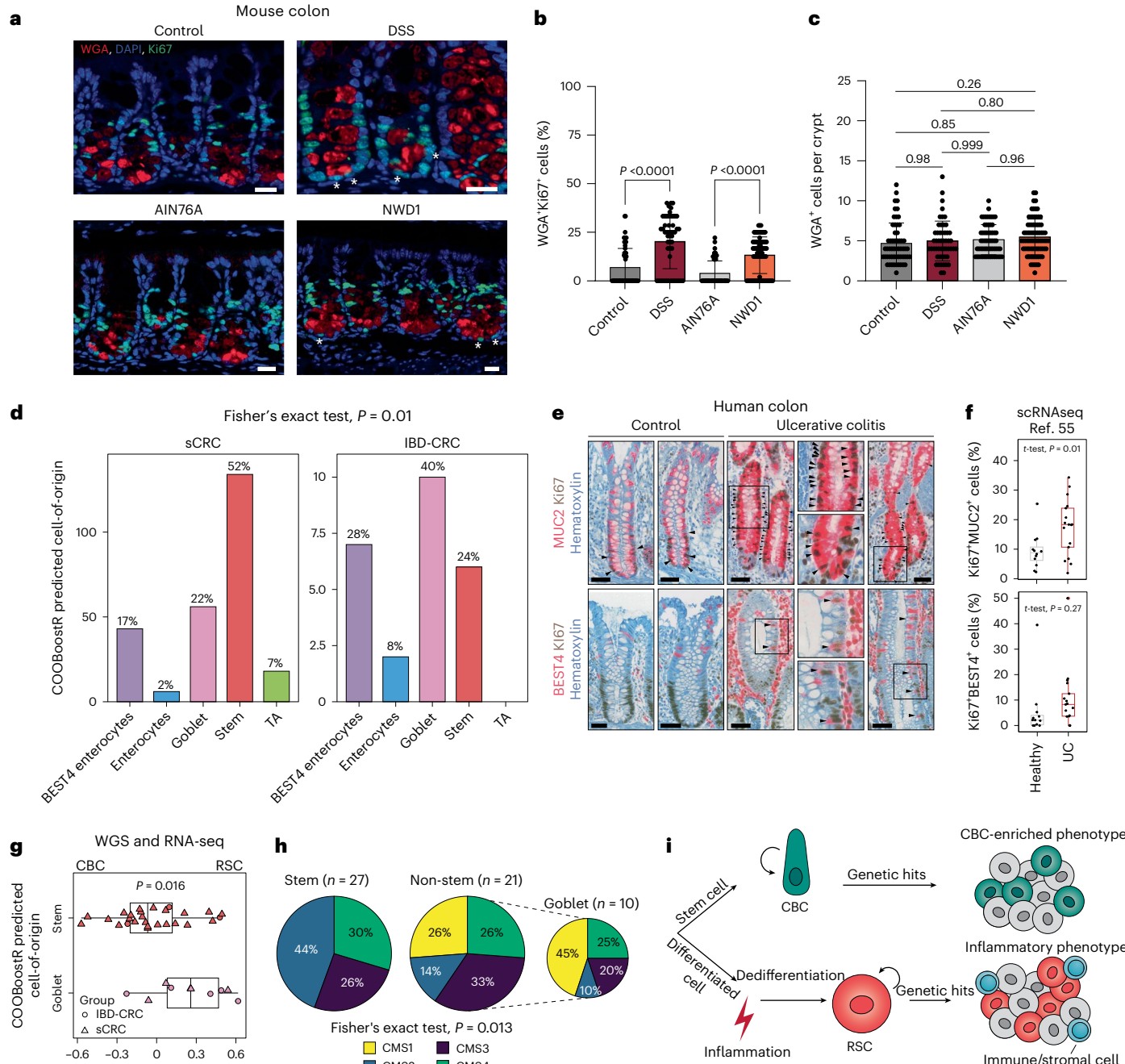

**Fig. 5 | Inflammation activates distinct cells-of-origin in the colon. a**, Colonic tissues from either untreated mice or mice administered 3% DSS for 7 days, as well as from mice fed AIN-76A or NWD1 synthetic diets for 3 months were analyzed for the presence of proliferative DCS cells. WGA was used to stain DCS cells[50], and Ki67 to mark proliferative cells. Tissues were counterstained by DAPI (nucleus). Asterisks mark WGA/Ki67 double-positive cells. **b,c**, Quantification of WGA+Ki67+ cells (**b**) and total WGA+ cells (**c**) in the lower colonic crypt of the mice, as shown in **a**. Scale bar, 20 μm. A minimum of 50 crypts from three different mice were analyzed. Data are presented as mean and s.d. *P* values denote two-tailed Tukey's tests for group comparisons. **d**, Bar plot showing the predicted cell-of-origin in IBD-CRC (*n* = 25) and sCRC (*n* = 257)[38] cohorts based on the COOBoostR computational approach[53] (Methods). The *P* value is the result of Fisher's exact test. **e**, MUC2, BEST4 and Ki67 IHC analysis of colonic tissues obtained from controls and patients with IBD. Asterisks indicate double-positive cells. Scale bars, 50 μm. **f**, Box plot showing percentage of cycling (Ki67+) MUC2+ and REG4+ cells in patients with ulcerative colitis (UC) and controls. scRNA-seq data are from ref. 55. Positivity was defined per cell by the presence of at least one read for that particular marker. Subsequently, cells were aggregated per patient to calculate

percentages. *n* = 12 healthy participants and *n* = 17 patients with ulcerative colitis. Box plots show the median, and lower and upper hinges correspond to the first and third quartiles. Whiskers extend from the hinges to maximum and minimum values, no further than 1.5× interquartile range. The *P* value shows the result of a two-sided *t*-test. **g**, Box plot denoting differences in the stem cell index based on stratification of predicted cell-of-origin in a subset of IBD-CRC and sCRC cases for which RNA-seq data were available (*n* = 27 stem, *n* = 10 goblet)[38]. The *P* value shows the result of a two-sided *t*-test. Box plots display the median, and lower and upper hinges corresponding to the first and third quartiles. Whiskers extend from the hinges to maximum and minimum values, no further than 1.5× interquartile range. **h**, Mapping of CMS on tumor samples stratified according to their predicted cell-of-origin. The *P* value shows the result of Fisher's exact test. **i**, Graphic abstract of the model arising from this study. Colon cancer can be initiated either from stem (ISC) or differentiated cells, the latter in response to inflammatory cues. RSC reprogramming is activated in support of the regenerative response. During this process, actively dividing RSCs expand the cell targets for tumor initiation and progression, leading to an alternative route to tumorigenesis earmarked by an inflammatory phenotype.

patient-derived colon cancers. More systematic and high-throughput screening approaches will likely lead to a novel stratification of the cases based on their cell-of-origin and specific etiologic factors such as western-style diets and chronic inflammation.

Overall, the increased colon cancer risk exerted by inflammation may reflect a 'trade-off' effect in which the chronic nature of the injury continuously stimulates dedifferentiation of committed lineages into stem-like cells in support of the tissue's regenerative response, thus resulting in enlargement of the pool of potential targets for tumor onset. Likewise, the same inflammation-inducing risk factors are bound to affect tumor progression towards malignancy thus resulting in a distinct group of colon cancers with a specific prognosis and response to therapy (Fig. 5i).

In our study, we used PC as a model of postmitotic and fully committed lineages and took advantage of the readily available *Lyz1*-Cre model[18]. However, the obtained results do not exclude a scenario in which lineages other than PCs can respond to the inflammatory insult by re-entering the cell cycle, and dedifferentiating and acquiring stem-like properties. In fact, the results obtained from machine-learning analysis of the whole-genome mutation spectrum of a large cohort of IBD-related and sporadic colon cancers revealed that multiple lineages are likely to actively contribute to tumor formation, possibly with a preference for secretory lineages (goblet cells). Moreover, as shown here and in a previous study[13], the observed ablation of *Lgr5*[+] ISCs in response to acute inflammatory stimuli is not complete and the persistence of some ISCs under inflammatory conditions makes their contribution to tumorigenesis in the context of inflammation possible. The latter would result in an increase in the intratumor heterogeneity of colon cancers originating from an inflammatory milieu and, accordingly, in an increase in therapeutic resistance and overall poor survival.

The putative participation of *Lgr5*[+] ISCs in tumor formation upon escaping the adverse effects of inflammation is also of relevance in view of the activation of the RSC state in this context. In their original study, Ayyaz et al.[30] reported the emergence of RSCs mainly upon irradiation, but also upon DSS administration. Because we did not detect RSCs in our single-cell profiling analysis, this is possibly explained by differences in the severity of the injury level. Complete RSC conversion may require high injury levels only attained through oncogenic targeting (as in the case of combined *Apc* and *Kras* mutations) or gamma-irradiation, whereas milder DSS- and diet-induced inflammatory insults elicit more subtle cellular perturbations. Of note, complete ISC ablation by diphtheria toxin results, even in the absence of DSS, in a partial RSC conversion[32], again underlining the relevance of the incomplete ISC depletion upon DSS in our study.

As shown in a previous study[59], ISCs can give rise to RSCs upon oncogenic targeting. Given the predominance of CBC-like profiles among ISC-derived adenomas, it is plausible that the identity of the cell-of-origin is a determinant of the efficiency or stability by which the RSC state is obtained.

Relative to the debate on the relative contribution of extrinsic risk factors versus the rate of stem cell division to cancer development[60,61], our results indicate that colon cancer etiologic factors such as inflammation and poor dietary habits are likely to result in quantitative and qualitative alterations in the stem cell niche, which ultimately predispose to neoplastic transformation. The resulting subset of tumors follow distinct evolutionary paths compared with WNT/MYC-driven LGR5-derived tumorigenesis, and are characterized by an inflammatory tumor phenotype more prone to infiltration from the TME. When compared with their LGR5 counterparts, PC-derived tumors (and those obtained by AOM/DSS) harbor a higher fraction of γδ[+] T cells versus T[reg] cells. Of note, the fraction of γδ2[+] T cells has been shown to be high in IBD and colitis-associated cancer[62], where they contribute to the inflammatory milieu by secreting cytokines such as IL-17 and tumor necrosis factor. The latter aligns with our findings where, in the tumors from

the AOM/DSS mouse model, γδ T cells are abundant and expressing *Il17a* and *Pdcd1*. This points to their putative role in the inflammatory milieu as observed for PC-derived tumors.

In conclusion, our results provide the basis for a new classification of colon cancer cases based not only on the cell-of-origin, but also on the underlying etiologic risk factors. Recent reports on the alarming increase in early-onset colon cancer cases may indicate yet another group of patients with distinct cells-of-origin as a result of specific lifestyle habits.

## Online content

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

## Methods

### Mice

The following inducible Cre strains were used: Lgr5[CreERT2-EGFP] (Jackson Laboratories, cat. no. 008875)[17] and pLys[CreERT2] (kind gift from Clevers lab)[18]. For lineage-tracing experiments, mice were crossed with R26[LSL-YFP] mice (Jackson Laboratories, cat. no. 006148) or R26[LSL-tdTomato] mice (Jackson Laboratories, cat. no. 007908). To target specific mutations in PCs, the above strains were further crossed with Apc[15lox] (Jackson Laboratories, cat. no. 029275)[14], Kras[LSL-G12D] (Jackson Laboratories, cat. no. 008179)[15] and Trp53[flox] (Jackson Laboratories, cat. no. 008462)[16]. For ablation experiments Lgr5[DTR-EGFP] mice[33] were bred with c-Kit[CreERT2] mice[65] and R26[LSL-tdTomato] mice (Jackson Laboratories, cat. no. 007908). C-recombinase was activated by intraperitoneal injections of tamoxifen (1 mg dissolved in 100% ethanol and subsequently in sunflower oil; Sigma, cat. nos. T5648 and S5007) once or three times in 4 days. Three days after tamoxifen injection, mice were treated once with diphtheria toxin to ablate Lgr5[+] cells. To induce acute intestinal inflammation, mice were administered 2–3% DSS in their drinking water for 7 days (MP Biomedicals, cat. no. 0216011050). DSS-driven inflammation and the corresponding mechanisms underlying the consequent PC dedifferentiation have been described in previous studies[13,66,67]. For diet experiments, mice were fed with AIN-76A or NWD1 diet[58] and collected at various time points (4 and 8 days, 3, 6 and 9 months). The formulation and composition of the AIN-76A and NWD1 diets are as outlined in ref. 46 (Research Diets). In brief, the western diets are based on the AIN-76A control diet. NWD1 was adjusted to provide higher fat and lower vitamin $D_3$, calcium donors to the single carbon pool and fiber that produces mouse consumption levels similar to those common in segments of western societies with a high incidence of CRC. All mice used here were inbred C57BL6/J.

For all experiments, mice were randomly assigned to experimental groups after matching for sex, age of 8–12 weeks and genotype. All protocols involving animals were approved by the Dutch Animal Experimental Committee and in accordance with the Code of Practice for Animal Experiments in Cancer Research established by the Netherlands Inspectorate for Health Protections, Commodities and Veterinary Public Health. Animals were bred and maintained in the Erasmus MC animal facility (EDC) under conventional specific pathogen-free conditions.

### Lineage tracing

Lys[CreERT2/R26LSL-YFP] mice were injected three times with tamoxifen (1 mg, intraperitoneally) on consecutive days. One week after the last tamoxifen injection, intestinal tissues were harvested for lineage-tracing analysis. Tissue samples were first dissected and washed with PBS buffer, and then fixed for 2 h at 25 °C with 4% buffered formaldehyde solution (Klinipath). Tissues were cryoprotected in 30% sucrose (Sigma) overnight at 4 °C, embedded in OCT (Optimal cutting temperature compound; KP cryo-compound, Klinipath), frozen on dry ice and sectioned at −20 °C. Tissues were cut into sections 4–8 μm thick. These sections were incubated in PBS containing Alexa 568 Phalloidin (1:100; Invitrogen) or Alexa 633 Phalloidin (1:100; Invitrogen) and 4,6-diamidino2-phenylindole (DAPI; Sigma) for 30 min at 25 °C, and washed in PBS-T buffer. Tissues were mounted in VECTASHIELD Mounting Medium (Vector Labs) and imaged with a LSM700 confocal microscope (Zeiss). Images were processed with ImageJ. Lineage-tracing frequency was quantified by dividing the number of intestinal crypt–villus axes containing YFP[+] ribbons (encompassing at least five YFP[+]-labeled cells) by the total number of counted YFP-labeled crypts. Lineage-tracing analysis was performed on three mice for each diet type; in each mice at least 60 YFP-labeled crypts were analyzed.

### Immunohistochemistry on mouse tissues

Tissues were fixed in 4% paraformaldehyde (PFA) overnight at 4 °C and embedded in paraffin. The 4-μm sections were dewaxed with xylene and hydrated in consecutive rounds of 70% and 100% ethanol. Antigen retrieval was performed in a 2100 Retriever pressure cooker (BioVendor) at pH 9 with Tris-EDTA buffer. After a 10-min incubation at room temperature with 3% hydrogen peroxidase, tissues were blocked with 5% skim milk powder (Millipore) in PBS-Tween. The following primary antibodies were used in an overnight incubation at 4 °C: β-catenin (BD Biosciences, cat. no. 610154); γ-H2AX (Cell Signaling, cat. no. 9718); green fluorescent protein (GFP; Thermo Fisher Scientific, cat. no. A-11122); Olfm4 (Cell Signaling, cat. no. D6Y5A); Lyz1 (Dako, cat. no. A0099); Dclk1 (Abcam, cat. no. ab37994); Muc2 (Santa Cruz, cat. no. sc-15334); ChgA (Novus Biologicals, cat. no. NB120-15160); E-cadherin (BD Biosciences, cat. no. 610182); Cd3 (Abcam, cat. no. ab5690); F4/80 (Cell Signaling, cat. no. D2S9R); and αSMA (Abcam, cat. no. ab5694). Slides were washed twice with PBS-Tween and incubated for 30 min at room temperature with the rabbit and mouse EnVision kits (Dako, cat. nos. K4001 and K4007). Slides were counterstained with hematoxylin, dehydrated with subsequent 70% and 100% ethanol, and mounted with Pertex (Histolab, cat. no. 00811). Whole slides were scanned with the Nanozoomer (Hamamatsu) and analyzed with NDP viewer v.2 (Hamamatsu).

### Immunohistochemistry analysis of human tissues

The study followed the guidelines of the European Network of Research Ethics Committees, in line with European, national and local regulations. As per national protocols, informed consent was not required for the immunohistological analysis of residual tissue material. Approval for the use of material from patients with IBD was obtained from the medical ethics committee of the Erasmus MC under license MEC-2009-041. Chromogenic double-labeling was performed on 4-μm thick whole-slide sections from formalin-fixed paraffin-embedded tissue blocks, on a validated and accredited automated slide stainer (Benchmark ULTRA System, Ventana Medical Systems) according to the manufacturer's instructions. Briefly, following deparaffinization and heat-induced antigen retrieval for 40 min at 97 °C, the tissue samples were incubated with KI67 (MIB-1 antibody, 0.4 μg ml$^{-1}$; Ventana, cat. no. 790-4286) for 32 min at 37 °C, followed by Ultraview detection (Ventana, cat. no. 760-500). After the stripping step, MUC2 (CCP58, 1:100; Dako, cat. no. M7313) or BEST4 (HPA058564, 1:100; Sigma-Aldrich) was incubated for 32 min at 37 °C and followed by detection with Ultraview Red (Ventana, cat. no. 760-501). Counterstaining was done using hematoxylin II for 12 min and a blue coloring reagent for 8 min. Each tissue slide contained a fragment of formalin-fixed paraffin-embedded pancreas or tonsil as an on-slide positive control.

### Immunofluorescence

For cryosections, tissue samples were first dissected and washed with PBS, then fixed for 2 h at 25 °C with 4% buffered formaldehyde solution (Klinipath). Tissues were cryoprotected in 30% sucrose (Sigma) overnight at 4 °C, embedded in OCT (KP cryo-compound; Klinipath), frozen on dry ice and sectioned at −20 °C. Tissues were cut into sections 4–8 μm thick. Immunofluorescence was performed using antibodies against OLFM4 (1:200; Cell Signaling Technologies), MKI67 (1:200; Novus Biologicals), LYZ1 (EC 3.2.1.17, 1:1,000; Dako) and KIT (D13A2, 1:100; Cell Signaling Technologies). Tissue sections were washed and blocked with 5% BSA in PBS-Tween for 1 h then incubated with the primary antibody overnight at 4 °C. Slides were washed twice with PBS-Tween and incubated for 2 h with goat anti-rabbit immunoglobulin G (H + L) secondary antibody, Alexa Fluor 488 conjugate (1:250; Life Technologies) and WGA 647 conjugate (1:500; Thermo Fisher Scientific). Tissues were counterstained with DAPI (Sigma) and, where indicated, with F-actin (Spirochrome, cat. no. SC001) for 30 min at 25 °C and washed in PBS-T. Tissues were mounted in VECTASHIELD HardSet Antifade Mounting Medium (Vector Labs) and imaged with a Zeiss LSM700 confocal microscope.

### In situ hybridization

In situ hybridization was performed using RNAscope Multiplex Fluorescent V2 Assay (ACDBio) technology in combination with the

RNA–Protein Co-Detection kit (ACDBio), according to the manufacturer's instructions. In brief, 4-µm paraffin-embedded sections were dehydrated, blocked for 10 min at room temperature with hydrogen peroxide and processed with 1× antigen retrieval co-detection buffer for 15 min at 99 °C. Sections were incubated with E-cadherin (1:250; BD Biosciences, cat. no. 610182) overnight at 4 °C. Hybridization was performed with a probe targeting mouse *Clu* (ACDBio, cat. no. 427891), and the probe signal was developed with Vivid dye 520 nm (1:1,500). Sections were counterstained by DAPI and anti-mouse Alexa 633 (1:250; Invitrogen) and mounted with ProLong gold mountant (Thermo Fisher Scientific). Samples were imaged on a Stellaris 5 confocal microscope (Leica) and images were processed with ImageJ.

### Image analysis

Images were scanned by Nanozoomer and imported in QuPath[68] as IHC images. Within intestinal tumors, regions of interest (ROI) were randomly assigned and consisted of 300–600 cells. ROIs were then processed with a positive cell selection plugin using default settings based on Cell DAB OD mean. The percentage of positive cells per ROI was exported and visualized in R, in which statistical tests were performed. Fluorescent images were processed with ImageJ.

### Quantification of Ki67/WGA cells in mouse colon

The percentage of Ki67$^+$WGA$^+$ double-positive cells was quantified by dividing the number of double-positive cells by the total amount of WGA$^+$ in the lower half of the colonic crypt. Only clearly detectable double-positive cells, as determined by the analysis of $z$-stacks, were counted. For each quantification, at least 30 crypts of at least three experimental mice were used.

### Quantitative PCR

RNA was converted to complementary DNA using the High Capacity RNA-to-cDNA kit (Applied Biosystems). Samples were run in duplicate on a 7500 Real Time system (Applied Biosystems). Quantitative PCR (qPCR) was performed using the TaqMan assay (Applied Biosystems) according to the manufacturer's instructions. Samples were normalized to the beta-actin (*Actb*) housekeeping gene. The gene-specific TaqMan probes were: Mm02619580_g1 (*Actb*), Mm00443610_m1 (*Axin2*), Mm00657323 (*Lyz1*) and Mm01545303 (*Dclk1*). Data were processed and visualized in R and statistical tests were used to assess significance with one-way analysis of variance (ANOVA) and Tukey's post hoc test.

### Isolation of mouse intestinal crypts and cell dissociation

Mouse small intestines were flushed with cold PBS, removed from fat and scraped with a glass slide to remove intestinal villi. Resected sample were then sectioned into 5–10-mm segments and washed with cold PBS before incubation for 30 min at 4 °C in cold PBS supplemented with 6 mM EDTA. After an additional PBS wash, crypts were detached from the muscle layer by four rounds of harsh pipetting with cold PBS. Crypts were treated for 10 min at room temperature with DNase (2,000 U ml$^{-1}$; Thermo Fisher Scientific) in Advanced DMEM/F-12 medium and passed through a 70-µm strainer. Purified crypts were dissociated with 1 ml of prewarmed TrypLE and DNase for 3 min at 37 °C. Single cells were pelleted in 10 ml of ADF (advanced DMEM F12), resuspended in 3 ml of 5% FCS HBSS and manually counted.

### Fluorescence-activated cell sorting

Cell suspensions were blocked with TruStain Fc blocking reagent (BioLegend) for 10 min at 4 °C. After washing with 5% FCS HBSS, cells were stained for 30 min at 4 °C in 100 µl with Lin antibodies CD31-BV421 (BD Biosciences, cat. no. 563356), CD45-BV421 (BD Biosciences, cat. no. 563890), and TER119-BV421 (BD Biosciences, cat. no. 563998)), CD24-APC (Sony Biotechnology, cat. no. 1109070) and cKit-PE (BioLegend, cat. no. 105808). For scRNA-seq experiments, the antibody mix was supplemented with 2 µl of hashing antibody (TotalSeq hashtag antibodies; BioLegend, cat. nos. 155831, 155833, 155837, 155843 and 155849) to label the cells with oligo barcodes. After three washes with 3.5 ml of 5% FCS HBSS, cells were filtered using a 40-µm strainer and resuspended in 1 ml of 5% FCS HBSS with 1 µg ml$^{-1}$ DAPI (Sigma-Aldrich, cat. no. D9542). FACS analysis and sorting were performed using FACSAria III (BD Biosciences). DAPI- and BV421-conjugated antibodies were detected using a 405 nm laser and 450/40 band-pass filter. GFP- and BB515-conjugated antibodies were detected using a 488 nm laser and 502 long-pass + 530/30 band-pass filters. tdTomato- and phycoerythrin (PE)-conjugated antibodies were detected using a 561 nm laser and 582/15 band-pass filter. Allophycocyanin (APC)-conjugated antibodies were detected using a 633 nm laser and 660/20 band-pass filter. Samples were sorted for epithelial (Lin$^{neg}$), Paneth-enriched (SSC$^{hi}$CD24$^{hi}$cKit$^{hi}$) and traced cells (YFP$^{hi}$/tdTomato$^{hi}$) according to previously established gates[11–13]. To optimize the sorting for single-cell experiments, the initial single-cell gate was followed by a more stringent, population-specific single-cell gate[11]. Analysis of FACS data was performed with FACSDiva (v.8.0.1). FACS strategy was visualized with FlowJo v.10.

### Organoid reconstitution assay

Whole crypts were extracted from the small intestine of *Lgr5*-EGFP mice fed for 3 months (from weaning) with either AIN-76A or NWD1 diets[58]. After cellular dissociation and FACS, single *Lgr5*$^+$ cells and PCs were sorted into separate Eppendorf LoBind Tubes. Reconstitution of *Lgr5*-EGFP$^{hi}$ stem cells with PCs (SSC$^{hi}$CD24$^{hi}$cKit$^{hi}$) was performed by co-pelleting sorted cells at 300$g$ for 5 min in Eppendorf LoBind Tubes, and by incubating them for 15 min at 25 °C as previously described[12,48]. The cells were then resuspended in 30 µl of Matrigel (Corning) and cultured in 96-well flat bottom dishes (Corning). In addition, Paneth and *Lgr5*$^+$ cells were plated singularly where indicated and cultured in standard ENR medium[69] supplemented with 10 µM Y-27632 (Sigma) and 1 mM Jagged-1 (AnaSpec). Cells were plated in triplicate and the generated organoids were counted 9 days after plating. Experiments were repeated at three independent moments.

### Droplet-based scRNA-seq

Sorted single cells were mixed in equal sample proportions with a pool of 25,000–50,000 cells in 5% FCS HBSS to a final concentration of ±1,000 cells µl$^{-1}$. The cell pool was recounted manually with Trypan blue and between 10,000 and 15,000 cells were loaded into a 10X Genomics Chip (Next GEM Chip G). Gel Beads-in Emulsion were generated with the Chromium Controller (10X Genomics) using Single Cell 3' v.3 chemistry with FeatureBarcoding. After generation of cDNA and a quality check with Bioanalyzer (Agilent), libraries were sequenced with the NovaSeq 6000 (Illumina) to a depth of 20,000–40,000 reads per cell. Raw data were processed with CellRanger count (cellranger-7.0.0)[70] and aligned to the mm10 mouse reference genome with the addition of the transgenes YFP and tdTomato.

### Analysis of scRNA-seq: inflammation and genetic targeting of PCs

For each single cell run, filtered gene–cell matrices were imported in R using Seurat (v.4.1.1)[71]. After a centered log ratio transformation of the hashtag oligo counts, cells were assigned back to their mouse-of-origin using HTODemux (positive quantile = 0.99). Cells assigned as doublets or negative for any of the barcodes were filtered from the data. Cells from mice of a different sex but the same genotype and experimental group were further separated based on the expression of *Xist*. After preprocessing, the five runs were merged into a single Seurat object. To remove putative doublets from the sorting, doublets were simulated and called by doubletfinder_v3 (nExp = 0.01)[72]. After removal of doublets, low-quality cells were removed by filters based on the percentage of mitochondrial genes (>10%), on nFeature_RNA (<200)

and nCount_RNA (<500). Next, data were integrated with the single-cell transform pipeline using reciprocal PCA dimension reduction to find integration anchors. Integrated data were further processed by dimension reduction with uniform manifold approximation and projection (UMAP) on the first 30 principal components. Cells were clustered using shared nearest neighbor (SNN) modularity optimization (resolution = 0.6) and annotated according to canonical marker genes. Traced cells were defined as a cell with a nonzero value for either tdTomato or YFP. Signatures were evaluated using the AddModuleScore function and visualized in experimental groups or cell clusters using violin plots. Pathway level evaluation was done using the GSVA package[73] on summarized expression profiles using average expression by cell cluster.

## Bulk RNA-seq of intestinal tumors

Intestinal tumors were macro-dissected from intestinal tissues and cut into 1–2 mm fragments that were then resuspended in 500 µl of TRIzol reagent (Ambion, cat. no. 15596026) for RNA isolation. RNA quality was first evaluated by Nanodrop ND-1000 (Thermo Fisher Scientific) and further purified with the TURBO DNA-free Kit (Invitrogen). Samples were sequenced with a read length of 150 bp (PE150) using the DNA nanoball (DNB-seq) protocol to a depth of 25 million reads per sample (BGI). Adapter trimming and sequencing quality control was performed with SOAPnuke software[74].

## Analysis of *Lyz1/Lgr5*-derived mouse tumors: bulk RNA-seq

FASTQ files were aligned with STAR (v.2.7.9a) to the mm10 mouse reference genome with Ensemble gene annotations[75]. After alignment, Sambamba (v.0.8.0) was applied to mark duplicates and perform flagstat quality checks[76]. Next, Subread (v.2.0.3) was used to count the primary strand-specific alignment[77]. Downstream analysis was performed in R using the DeSeq2 package (v.1.34.0)[78]. In brief, counts were log2 normalized and used for PCA on the top 500 variable expressed genes. Differential expression analysis was done following the DeSeq pipeline with negative binomial generalized linear model fitting and Wald statistics, and genes were filtered using log2-transformed fold change > 1.5, $P_{adj}$ < 0.01. GSEA was done with the fgsea package (v.1.20.0) using the HallMarks gene modules from the molecular signature database[39]. GSVA was performed using the GSVA package (v.1.42.0)[73]. The relative proportion of RSC/CBC stem cell types was computed with the ISC index using default settings[31].

## Analysis of PCs upon exposure to inflammation and western-style diet: bulk RNA-seq

FASTQ files were removed from TrueSeq adapters using Trimmomatic (v.0.36). STAR[75] was used to align the reads to the mm10 reference genome using Gencode annotation release M15 (GRCm38.p5). Mapping quality was assessed with Sambamba Flagstat statistics. Count files were generated using FeatureCounts (Subread) and further processed in R using the DeSeq2 package. Differential expression analysis was performed between the groups 'NWD1 diet' (n = 3) and 'AIN-76 diet' (n = 3) with the Wald test. A heatmap was produced using z-score scaled data and visualized with the ComplexHeatmap package (v.2.12.1)[79].

## Analysis of IBD-CRC and sCRC RNA-seq cohort

RNA-seq material was previously described[38] In brief, 64 CRCs entered HiSeqLncRNA-Seq library preparation and paired-end sequencing using Illumina HiSeqXTen. Salmon (v.0.12.0; quant mode with ValidateMappings) was used to map raw sequences onto the human transcriptome (Ensembl release 79). Gene-level quantification was done with DESeq2 (v.1.18.1) followed by a variance-stabilizing transformation and Limma (v.3.34.9) correction of sequencing batch effects[78]. Only MSS tumors were included in the analyses. Differential gene expression was calculated as an ANOVA between 38 MSS sCRCs and 14 MSS IBD-CRCs. GSVA (v.1.42.0)[73] analyses were run using default options (Gaussian kernel) and predefined gene sets from the molecular signature database[39].

Consensus molecular subtypes were mapped to the samples with the random forest classifier of CMSClassifier R package.

## Cell-of-origin predictions from IBD-CRCs and sCRCs

**Processing of whole-genome sequencing data of CRCs.** DNA was isolated from fresh-frozen tumor, normal colon or blood. Paired tumor-normal whole-genome sequencing (WGS) data were generated from a total of 25 MSS IBD-CRCs and 257 MSS sCRCs. Libraries were prepared using TruSeq Nano DNA HT or TruSeq PCR-Free Kit (Illumina), followed by paired-end sequencing using the Illumina platform (HiSeqXTen/HiSeq2000). More details are given in ref. 38. Sequence alignment to the GRCh38 reference genome, other data preprocessing steps and somatic variant calling were performed with GenomeAnalysisToolkit GATK4 best practices workflow (v.4.0.4.0)[80]. Mutations were filtered based on allele fraction to extract clonal somatic variants. Genome-wide mean and standard deviation values were calculated for each tumor; somatic variants were then filtered to a minimum allele fraction of one standard deviation below the mean. On average, 85% of variants per tumor (interquartile range 83–87%) were kept-in as clonal somatic variants for the subsequent analysis. Similar to previous work[51,52], the human reference genome (hg38) was processed in nonoverlapping windows of 1 million base pairs. We excluded window-regions with <92% of bases within uniquely mapped 36-base polymer/oligonucleotides, regions overlapping telomeres and centromeres, and regions outside autosomes. Altogether 2,387 windows remained for the analysis. To identify differentially mutated windows, patients were normalized by the average of their windows and IBD-CRCs versus sCRCs were compared with an Wilcoxon rank sum test (P < 0.01).

**Processing of scATAC data from human colon.** Normal colon (n = 9) samples from ref. 54 were obtained from the public domain under GEO identifier GSE201349. Signac[81] was used for processing and the FeatureMatrix function was used to generate a count matrix based on the previously described 2,387 windows. Cell labels were retrieved from the original publication and reduced to six major classes by the aggregation of clusters from the same cell type. To generate the embedding, batch correction was performed with IntegrateData, and UMAP was computed on the 2:30 latent semantic indexing components. To generate the chromatin mark features per cell type, counts from the scATAC dataset were grouped with the AggregateExpression function and equalized to 10 million reads per cell type.

**Predictions with COOBoostR.** COOBoostR[53], an extreme gradient boosting-based (XGBoost) machine learning algorithm designed for the prediction of cells-of-origin, was used to predict the putative cells-of-origin for each individual sample from the WGS cohort (n = 282 tumors) based on epigenetic profiles from the human colonic scATAC data. We ran the algorithm with default values from XGBoost for the learning rate (mEta ()= 0.3) and the maximum depth of the trees (mdepth = 6). Tumor predictions were aggregated on a cohort level, and IBD-CRC/sCRC were compared with Fisher's exact test. WGS and RNA-seq data were available for a subset of patients (n = 48 tumors). Hence, for these tumors, CMS labels and signatures from the GSVA were compared based on stratification by putative cell-of-origin.

## Analysis of human CRC bulk RNA-seq cohort

The log2 normalized expression of the meta cohort described in ref. 41 was obtained from the R2 Genomics Analysis and Visualization Platform and subsequently processed in R. GSVA was performed with the GSVA package using the Gaussian kernel[73]. Subsequently, correlations of gene signatures (Supplementary Table 4) were visualized with ggcorr using Pearson correlation. Next, samples were classified as colitis-like, sporadic-like or intermediate based on their GSVA scores for the IBD-CRC and sCRC signatures. Using this classification, samples

were visualized with their annotated CMS. Survival analysis was performed with the survival and survminer packages in R. Multivariate analysis was performed with the log-rank test and pairwise comparisons were fitted using a Cox proportional hazards regression model.

## Analysis of human scRNA-seq cohorts

**Colorectal cancer dataset.** Filtered count matrices were retrieved from the gene expression omnibus (KUL cohort: GSE144735; SMC cohort: GSE132465)[40] and processed in R with Seurat. After normalization and scaling of the Seurat object, gene expression was summarized per cell type using AverageExpression based on the overlapping differentially expressed genes of mouse (Paneth versus *Lgr5*) and human (IBD-CRC versus sCRC) tumor groups. The average expression was *z*-scored and visualized as a heatmap using the ComplexHeatmap package[79].

**Ulcerative colitis dataset.** Data from ref. 55 were processed with the Seurat[71] workflow. A subset was taken encompassing the epithelial cells according to annotation from the authors. Cells were defined as positive when they had nonzero counts for that particular gene. Cells were aggregated on patient level and a *t*-test was performed to compare number of double and single positive cells across the ulcerative colitis (*n* = 17) and control (*n* = 12) groups.

## Analysis of mouse intestinal single-cell datasets

**Homeostasis and gamma-irradiation.** Count matrices from refs. 64 and 30 were downloaded from GEO using GSE92332 and GSE117783, respectively. After preprocessing using the Seurat workflow, Seurat objects were merged and gene expression was averaged by cell type and study using AverageExpression. Next, GSVA[73] was computed using the Gaussian kernel and the output was visualized in a heatmap using the ComplexHeatmap package[79].

**AIN-76A and NWD1 diets: scRNA-seq.** The intestinal epithelia of mice fed control (AIN-76A) and western-style diets (4 days NWD1, 4 days NWD1 plus 4 days AIN recovery) were collected and sent for 10X Genomics scRNA-seq analysis (*n* = 3 mice per group)[47]. After preprocessing with Seurat and batch correction using the integrated assay workflow, the Seurat clusters were annotated according to canonical markers. CytoTRACE[49] was computed on the whole dataset using default settings. Next, PCs (*n* = 991 cells) were taken from the dataset, subclustered and visualized by UMAP with a split by the dietary condition. The DSS signature was computed using AddModuleScore and projected onto the UMAP embedding. The Paneth subcluster that appeared in the NWD1 conditions and was associated with the DSS signature was relabeled as 'diet response'. Next, marker genes were visualized using violin plot based on the separation of diet response PCs and other PCs. Last, PCs from the diet dataset were merged with PCs from the DSS dataset. Average expression was computed based on experimental condition, and GSVA analysis (Gaussian kernel) was performed using the predefined gene sets from the Molecular Signature Database (Hallmark set)[39,73]. A filter was applied to select pathways that were both upregulated in DSS and NWD1 condition (ΔGSVA = 0.2). The output was visualized as heatmap using the ComplexHeatmap package[79].

**AIN-76A and NWD1 diets: scATAC sequencing.** The intestinal epithelia of mice fed with control (AIN-76A) and western-style diets (3 months) were collected and sent for 10X Genomics scATAC sequencing analysis (*n* = 2 mice per group)[47]. Processing was done with Signac[81], and UMAP embedding was produced based on the 2:30 latent semantic indexing components. Unsupervised clustering was performed by a SNN modularity optimization-based clustering algorithm (algorithm: 1, resolution: 1.2). Cell-type prediction scores were obtained by integration with the AIN-76A/NWD1 scRNA-seq dataset using the FindTransferAnchors and TransferData functions. Two of twenty-one clusters had high association with the PC prediction score

(clusters 5 and 20). PCs were defined as part of either cluster 5 or cluster 20 with the additional condition of a nonzero prediction score for PCs. Next, PCs (*n* = 945 cells) were taken from the dataset, and visualized by UMAP with a split by dietary condition. Cluster 5 represented cells from both diets, whereas cluster 20 was enriched in NWD1 and therefore relabeled as 'diet response'. To identify differentially accessible peaks, FindMarkers was used with a logistic regression test. Peaks were filtered based on *P* < 0.05, visualized with a RegionHeatmap and RIdeogram on the mouse genome, and gene ontology was performed on the closest genes using the EnrichR package.

**Lgr5 ablation with diphtheria toxin.** Count matrices were obtained from GEO (GSE183299) and processing was done using the Seurat workflow[71]. The RSC signature was evaluated using the AddModuleScore function and visualized across experimental groups and cell clusters using the ComplexHeatmap package[79]. A subset was made of ISCs, and differential expression analysis was performed with FindMarkers. The results were visualized with the EnhancedVolcano package, where marker genes of PCs were highlighted.

**AOM/DSS-derived tumors.** Data from ref. 35 were downloaded through GEO (GSE134255) and processed with Seurat[71]. Differential abundance analysis was performed with MiloR[82] workflow using default parameters. The RSC signature and PC cluster 4 signature were evaluated using the AddModuleScore function and visualized with violin plots. To investigate T cell heterogeneity, a subset of the T cells was taken, dimension reduction was performed with UMAP (dims = 1:30, spread = 0.5) and cells were clustered with SNN modularity optimization (Leiden method, resolution = 0.1).

## CIBERSORTx analysis of AOM/DSS adenomas

CIBERSORTx[37] was leveraged to predict the fractions of various cell types in tumors arising in the AOM/DSS context. Count data from AOM/DSS-derived adenomas from ref. 34 (GSE178145) were converted to counts per million using Edger. Single-cell count data from PC upon genetic targeting were provided as a reference and converted to a signature matrix in CIBERSORTx using default parameters. For cell-type imputation, CIBERSORTx was applied in relative mode using 100 permutations.

## Immune profiling

Immune cell fractions were predicted from tumor bulk transcriptomic profiles with CIBERSORTx in absolute mode using 100 permutations and batch correction B[37]. Immune cell fractions were estimated with the LM22 (ref. 83) and ImmuCC[84] signature matrices for the human and mouse samples, respectively. For visualization, the estimated fractions were *z*-scored, averaged by clinical or experimental group, and displayed as a bar plot or heatmap using the ComplexHeatmap package[79]. Repertoire analyses were performed with MiXCR[85] using the analyze rna-seq function with default parameters. Immune receptor clonotypes were quantified using the immunarch package with the repExplore function and the volume method.

## Statistics and reproducibility

Statistical tests were performed in R v.4.3.2 with the ggpubr and rstatix packages. Representative micrographs come from at least *n* = 3 biologically independent samples/animals, and illustrate results that were observed consistently throughout a series of independent experiments. For the cell-of-origin experiments, no statistical method was used to predetermine sample size as effect sizes were unknown. Animals were randomly assigned to experimental groups.

## Reporting summary

Further information on research design is available in the Nature Portfolio Reporting Summary linked to this article.

## Data availability

All data relevant to this study are made available. Transcriptomic sequencing data has been deposited in the Gene Expression Omnibus (GEO), and is available using the following identifiers: GSE221819 (bulk RNA sequencing of murine tumors); GSE221820 (single-cell RNA sequencing of genetically targeted Paneth cells); GSE221818 (bulk RNA sequencing of sorted Paneth cells treated with control and western-style diet). Additional datasets referenced in this study are publicly available in the Synapse and GEO repositories: previously published scRNA-seq studies relating to the mouse small intestine in homeostasis were also used (Haber et al.[64], GSE92332), upon irradiation (Ayyaz et al.[30], GSE117783), *Lgr5* ablation (Singh et al.[32] GSE183299) and upon feeding with western-style diet (Choi et al.[47], scRNA-seq: GSE188577, scATAC sequencing: GSE228006). Mouse colonic scRNA-seq data from AOM/DSS tumors was obtained from Vega et al.[35] (GSE134255) and bulk RNA-seq was from Chen et al.[34] (GSE178145). Human CRC bulk RNA-seq and scRNA-seq studies were retrieved from Guinney et al.[41] from Synapse (syn2623706), Rajamäki et al.[38] and Lee et al.[40] (KUL cohort: GSE144735; SMC cohort: GSE132465), respectively. Human scRNA-seq data from patients with ulcerative colitis (Smillie et al.[55]) was downloaded from the Broad DUOS platform (duos.broadinstitute.org) after a data transfer agreement. Single-cell ATAC data from human colon was obtained from Becker et al.[54] (GSE201349). Source data are provided with this paper.

## Code availability

The software used in this study is publicly available. Libraries were used according to their vignettes with parameter descriptions as specified in the Methods and versions as depicted in the Reporting Summary. Custom code is available at GitHub (github.com/mpverhagen/NatGen-24).

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

## Acknowledgements

We thank E. Bindels and R. Hoogenboezem for their support to generate the single-cell sequencing data, J. D. Windster for help with in situ hybridizations, W. S. van de Geer for analysis support of bulk sequencing data; W. Szymanska, E. Middendorp Guerra and E. de Vet for help with the IHC and scRNA-seq analyses; the PARTS research facility for help with tissue processing and IHC; V. H. A. E. Verhagen for discussions on XGBoost machine learning; D. Kortleve and R. Debets for input on tumor immunology; F. Koning and N. F. C. C. de Miranda for their advice regarding γδ T cells; B. Vogelstein for helpful discussions. We thank F. de Sauvage and Genentech for kindly providing the Lgr5^DTR-EGFP mouse model and D. Saur for kindly providing the c-Kit^CreERT2 mice. This study was financially supported by the Dutch Cancer Society (project no. 11407), and in part by the National Cancer Institute (project nos. R01CA214625, R01CA229216 and P30-013330), the National Institute on Aging (project no. P30 AG038072, 5T32AG023475-20), Academy of Finland (Finnish Center of Excellence Program 2018–2025 312041, Academy Professor grant nos. 319083 and 320149), iCAN Digital Precision Cancer Medicine Flagship (grant no. 320185), Cancer Society of Finland, Sigrid Jusélius Foundation (grant no. 220002) and the Jane and Aatos Erkko Foundation. Funding for grants IIG_2014_1181 and IIG_FULL_2022_015 was obtained from Wereld Kanker Onderzoek Fonds as part of the World Cancer Research Fund International grant program.

## Author contributions

M.P.V., R.J., M.S. and A.S. designed and performed experiments. M.P.V., J.C., N.V., L.A.A. and L.H.A. generated and analyzed sequencing datasets. K.R., P.P., S.S., B.v.d.S. and T.P.P.v.d.B. performed immunohistochemistry. D.S., M.D., and A.C.d.V. were responsible for pathological examination and IBD patient selection. M.P.V. and R.F. wrote the manuscript. R.F. supervised the study and was responsible for the concept and design of the study.

## Competing interests

The authors declare no competing interests.

## Additional information

**Extended data** is available for this paper at https://doi.org/10.1038/s41588-024-01801-y.

**Correspondence and requests for materials** should be addressed to Riccardo Fodde.

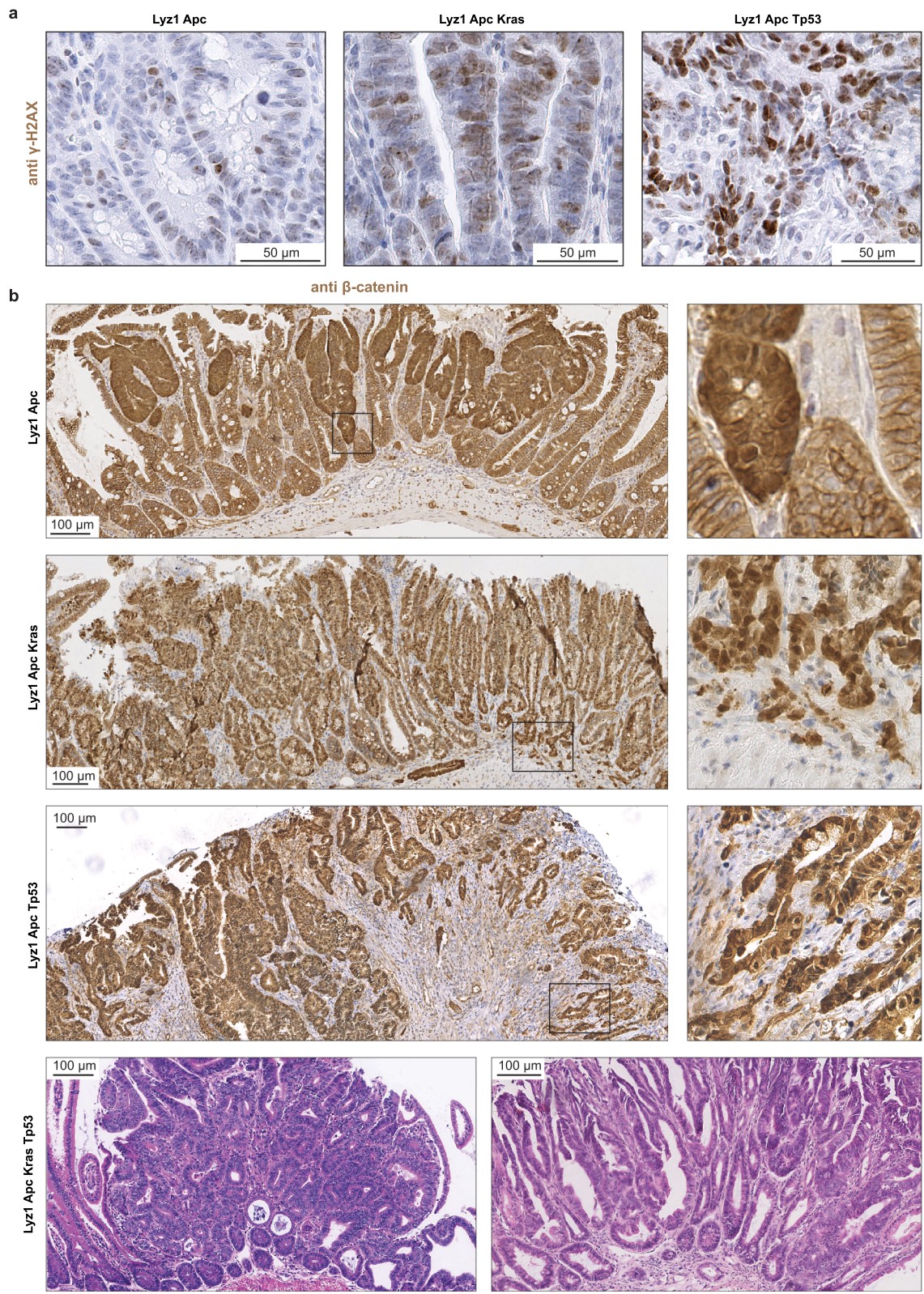

**Extended Data Fig. 1 | Immuno-histologic analysis of Paneth-derived tumors.**
**a**. Representative IHC images of Paneth-derived tumors from different genetic backgrounds, stained with phospho-histone H2A.X (Ser 139). **b**. Top three panels: IHC analysis of Paneth-derived intestinal tumors with different genetic backgrounds (as indicated), stained for β-catenin. Bottom: hematoxylin and eosin histological stains of Paneth-derived tumors from Lyz1/Apc/Kras/Tp53 mice.

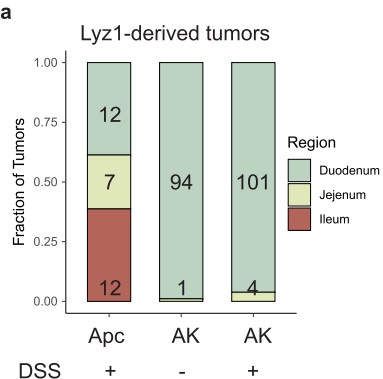

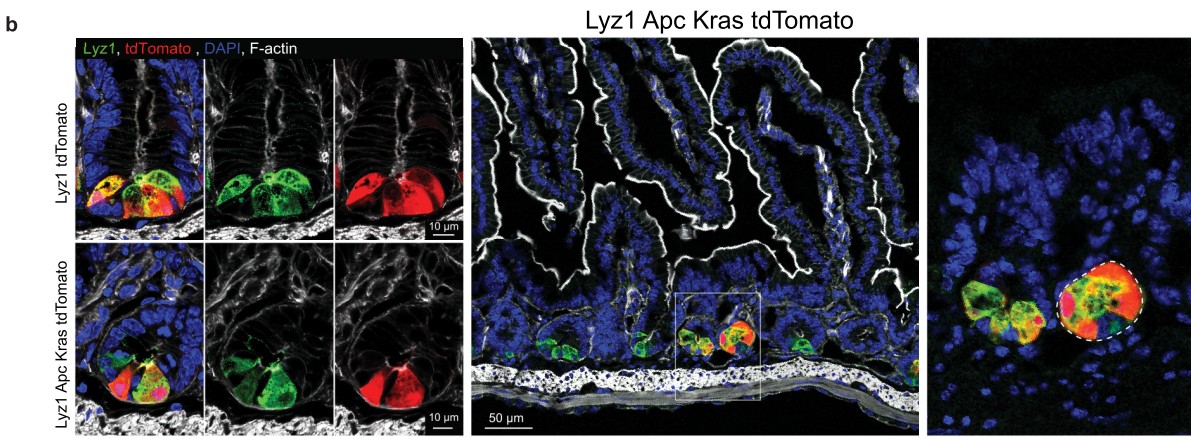

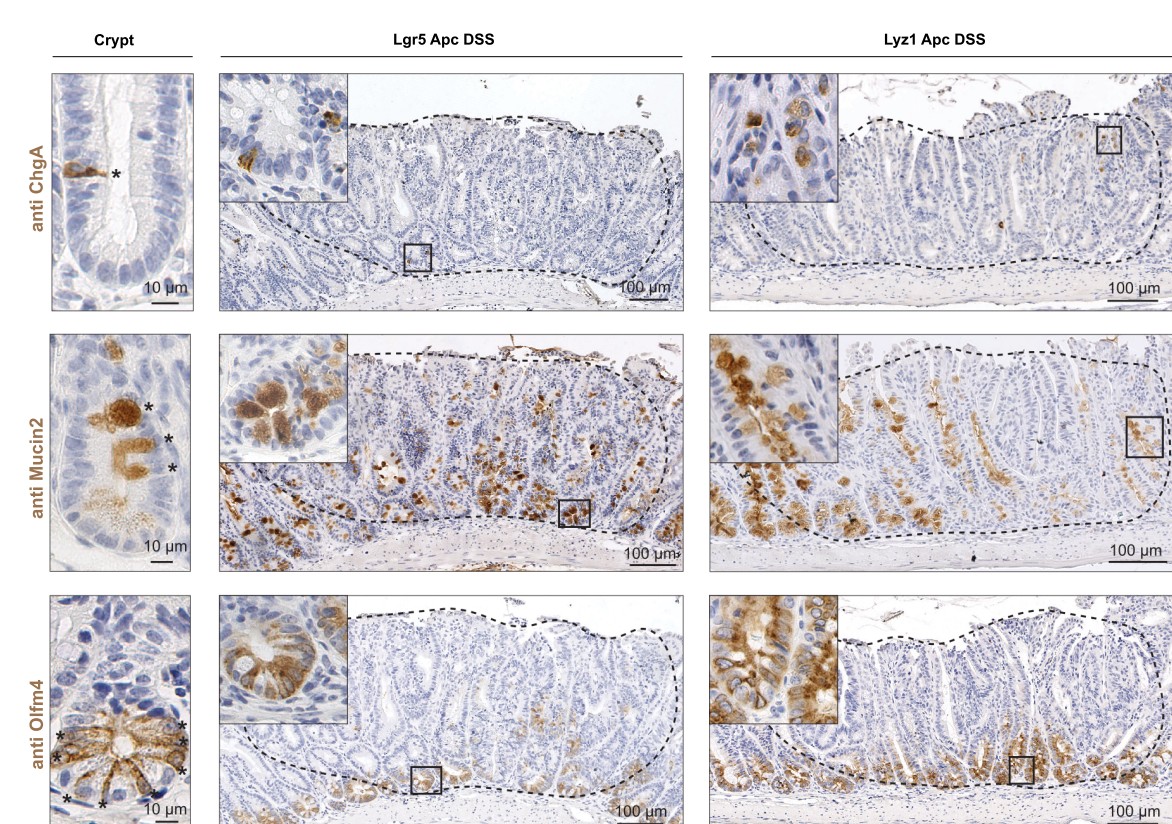

**Extended Data Fig. 2 | Characterization of PC-derived small intestinal adenomas. a**. Stacked bar plot displaying the localization and distribution of PC-derived adenomas across the small intestine. **b**. Immunofluorescence imaging of intestinal crypts from Lyz1/tdTomato and Lyz1-AK/tdTomato mice. **c**. IHC analysis of the lineage-specific ChgA, Muc2, and Olfm4 markers in PC- and Lgr5-derived adenomas.

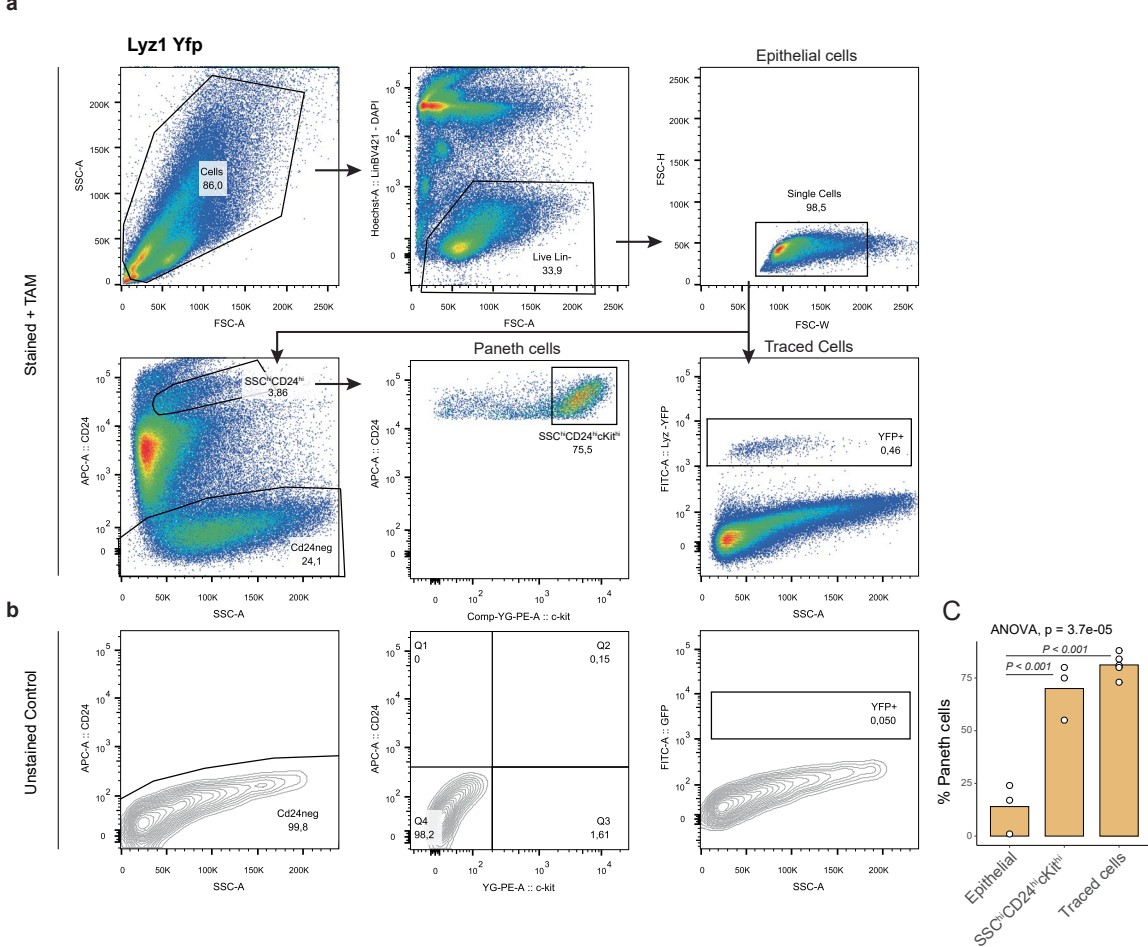

**Extended Data Fig. 3 | FACS gating strategy for Paneth cells enrichment and isolation. a**. FACS gating strategy established for the isolation of live, single, epithelial cells (top). Subsequently, Paneth cells were enriched using the SSChiCD24hi gate and further purified based on high cKit levels (SSChiCD24hicKithi). Alternatively, traced cells were sorted based on positive expression for Yfp or td-Tomato when the appropriate reporter mice were employed. **b**. Negative control employed to establish reference levels based on unstained samples. **c**. Bar plot of the percentage of Paneth cells obtained by the distinct gating strategies. N = 3 biologically independent experiments, and N = 5 for the tracing experiments. P values depict results of one-way ANOVA and Tukey tests. Data are presented as mean values +/- SD.

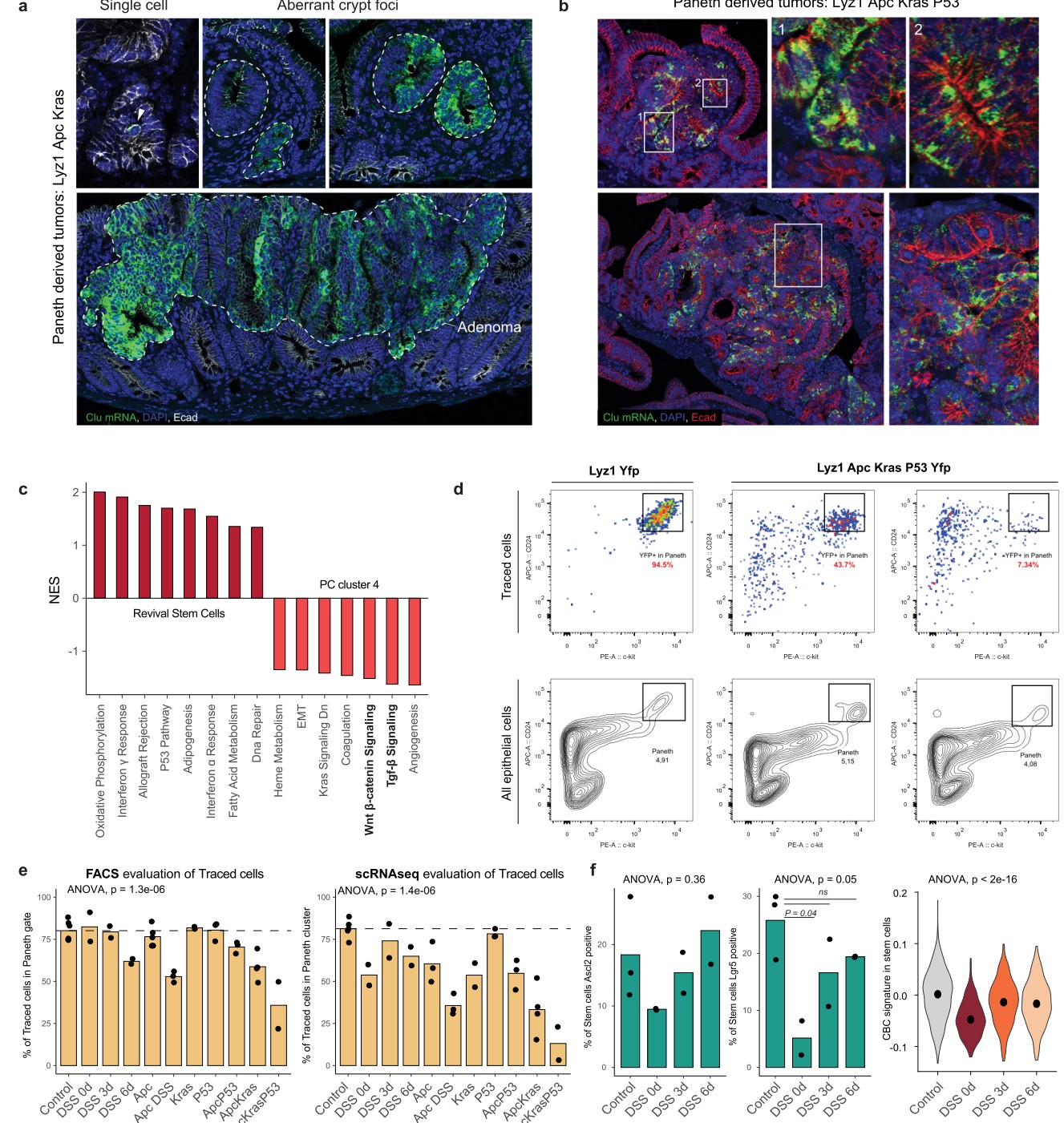

**Extended Data Fig. 4 | Paneth cells dedifferentiate upon DSS-driven inflammation. a**, **b**. In situ hybridization (ISH) analysis of Clu mRNA (green) expression at various stages during PC-derived tumorigenesis in Lyz1-Apc/Kras and (b) Lyz1-Apc/Kras/Tp53 mice. **c**. Bar plot showing filtered pathways (Pval < 0.05, abs NES > 0.5) of the gene set enrichment analysis comparing RSCs with PC Cluster 4. **d**. FACS analysis of lineage-traced (Yfp + ) cells in wild type (Lyz1-Yfp) and AKP-mutant (Lyz1-Apc/Kras/Tp53) mice. The latter clearly localize outside of the CD24hicKithi gate where wild type PCs reside. **e**. Bar plot relative to the

percentages of lineage-traced cells evaluated by FACS using the CD24hicKithi PC-specific gate (left), or by scRNAseq analysis (% within Paneth cluster; right). P values denote significance of one-way ANOVA. **f**. Left: Bar plots showing the average percentage of Ascl2+ and Lgr5+ stem cells in control and DSS-treated animals. Right: Violin plots showing a decrease in the crypt base columnar (CBC) signature upon DSS treatment. P values denote significance of one-way ANOVA and Tukey test for group comparisons.

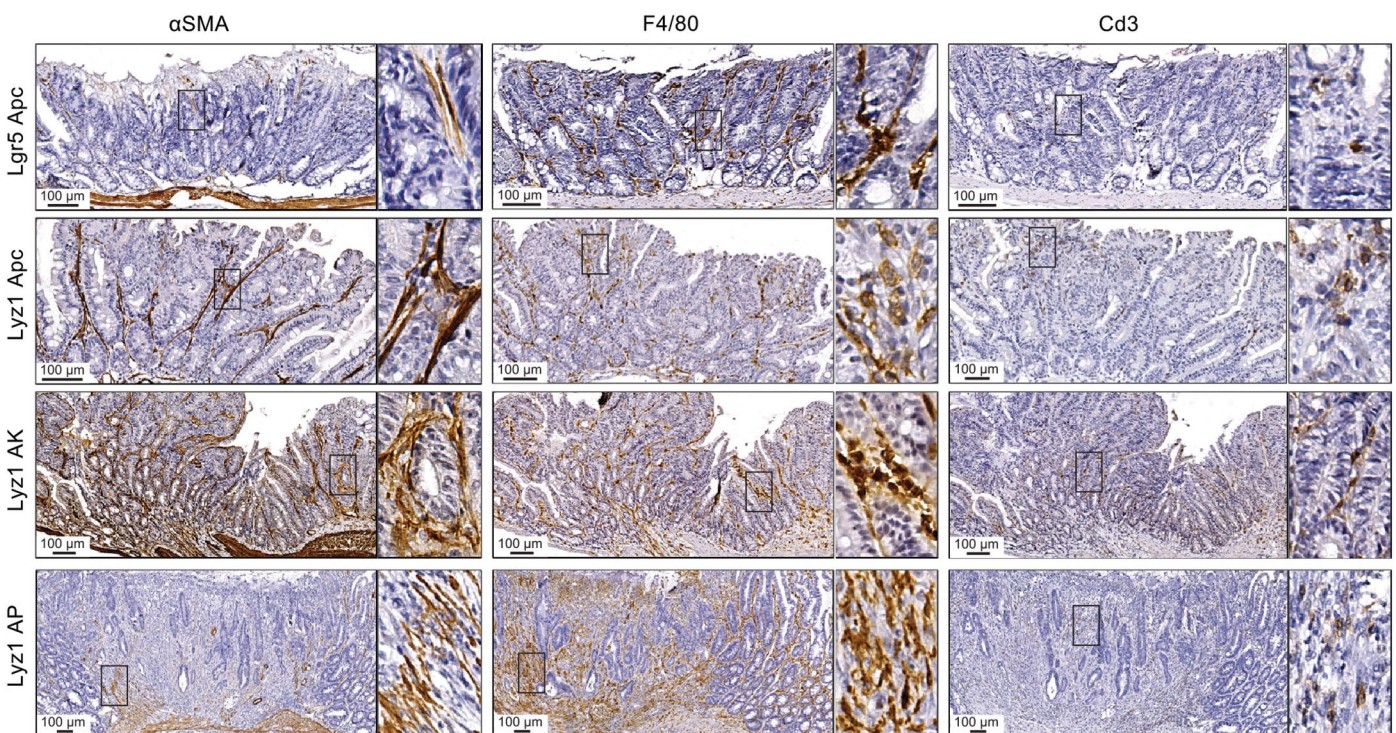

**Extended Data Fig. 5 | Distinct TME features in CBC- and PC-derived small intestinal adenomas.** IHC analysis of Lgr5+ and Lyz-derived adenomas with the tumor microenvironment markers αSMA (fibroblasts), F4/80 (macrophages), and Cd3 (T cells).

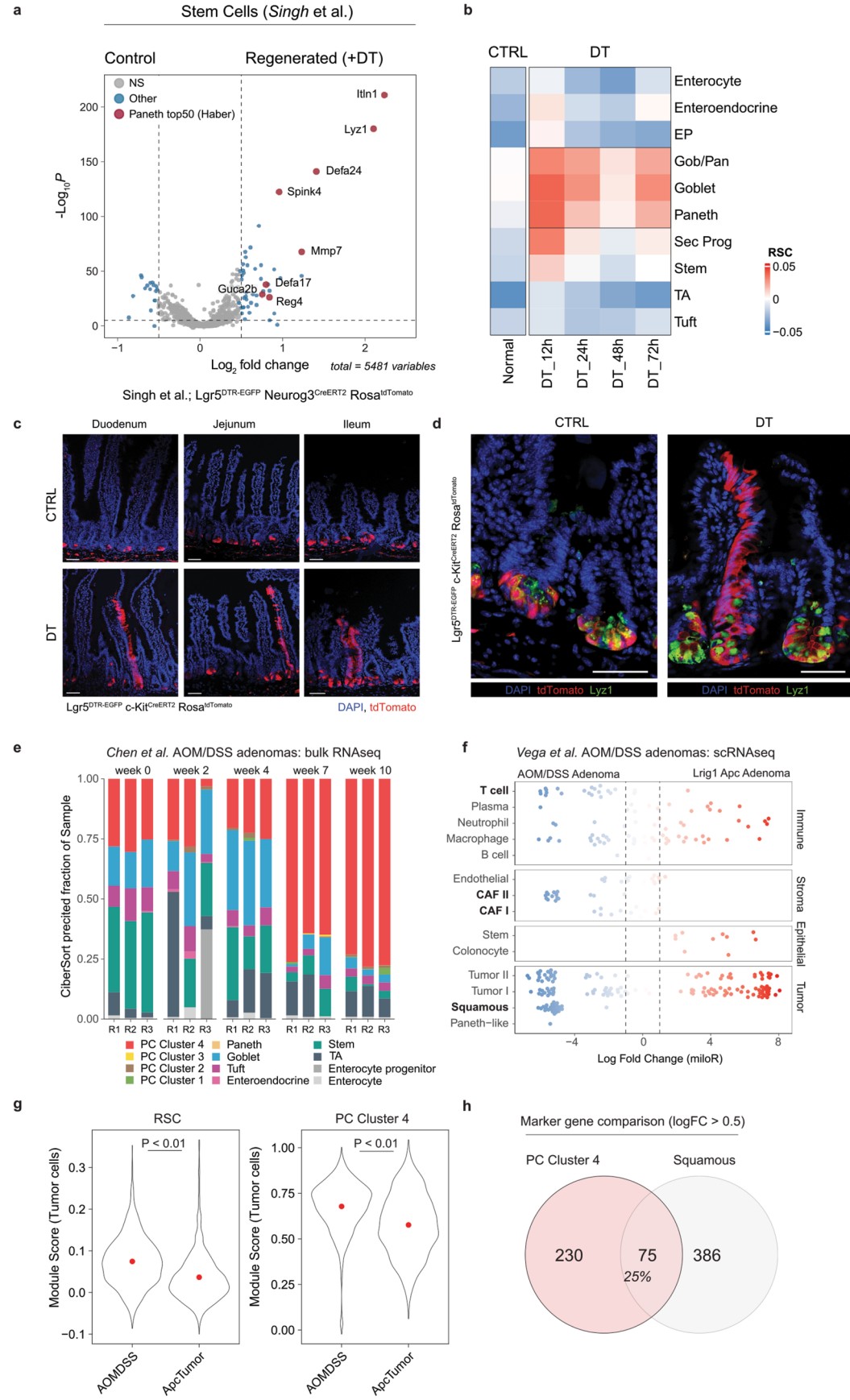

**Extended Data Fig. 6 | See next page for caption.**

**Extended Data Fig. 6 | _Lgr5_[+] CBCs ablation triggers dedifferentiation of Paneth and goblet secretory lineages. a**. Differential expression analysis of intestinal stem cells in homeostasis and during regeneration following Lgr5[+] CBCs ablation. scRNAseq are from Singh et al.[30]. Several among the top 50 differentially expressed genes in the regenerated ISCs are Paneth-specific markers (red). Pvalues denote the result of the Wilcoxon Rank Sum test. **b**. Heatmap showing transcriptional activity of the RSC program upon Lgr5[+] CBCs ablation. **c**, **d**. Lineage tracing of DT-treated Lgr5[DTR-EGFP];c-Kit[CreERT2]; Rosa[tdTomato] mice across different intestinal sections (**c**.) and double stained with Lyz1/tdTomato (**d**.). Scale bar = 50 µM, 2 mice per group were analyzed. **e**. Stacked bar plot of the cell type fractions in AOM/DSS adenomas as predicted by Cibersort analysis[35] (see Methods). Bulk RNAseq data are from Chen et al.[32]. **f**. Differential cell type abundance (MiloR63; see Methods) between AOM/DSS and Lrig1-Apc adenomas. scRNAseq data are from Vega et al.[33]. **g**. Comparison of RSC and PC Cluster 4 module score in tumor cells from AOM/DSS and Lrig1-Apc adenomas. P values depict significance values of two-sided Wilcoxon test. **h**. Venn diagram indicating the overlap between marker genes from PC Cluster 4 and the AOM/DSS-specific squamous tumor cell subpopulation.

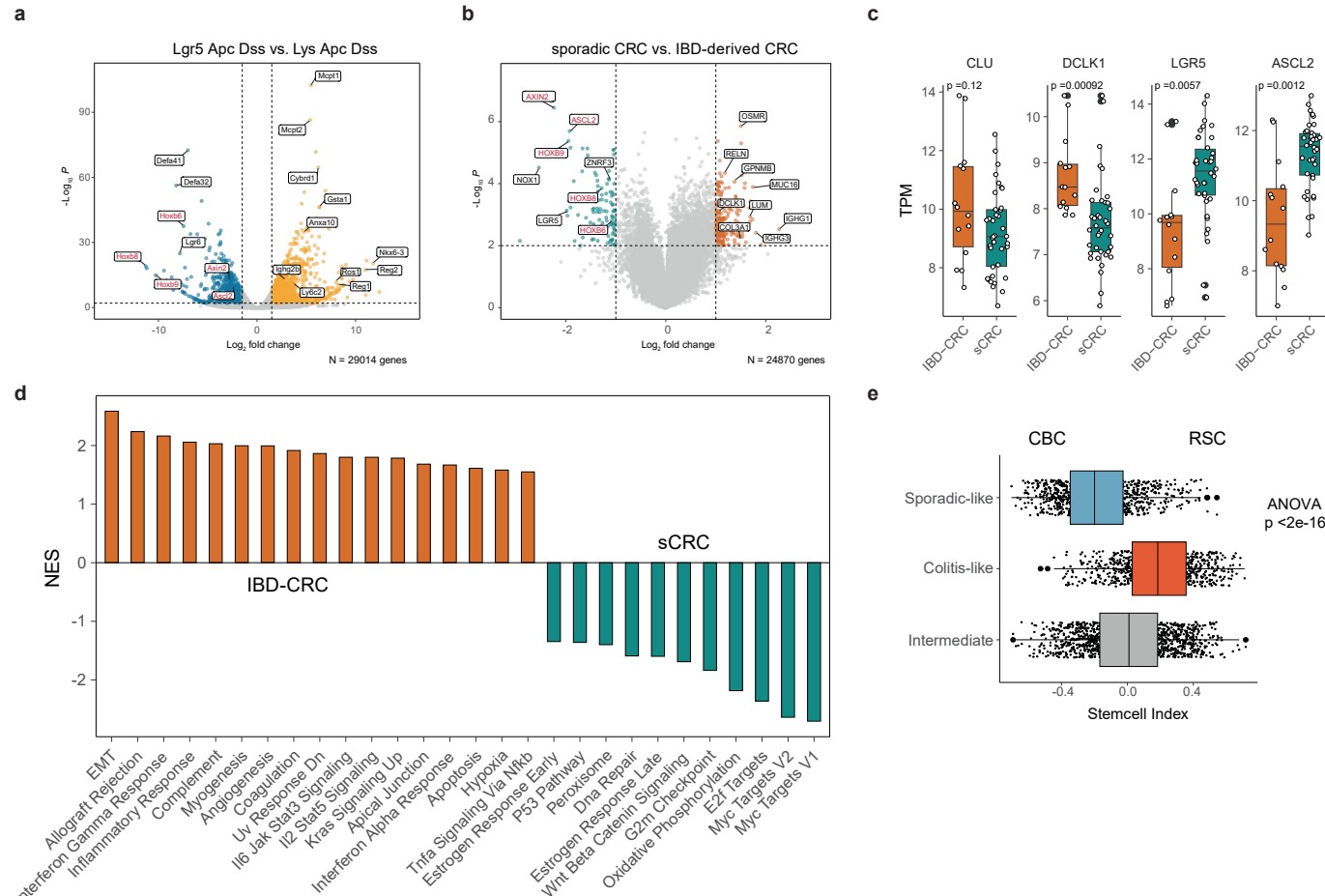

**Extended Data Fig. 7 | Differential gene expression analysis between IBD colon cancers and murine PC-derived small intestinal adenomas. a**. Volcano plot showing differentially expressed (Pval < 0.01, FC cutoff 1.5) genes between Lgr5- and Paneth-derived intestinal tumors. Two-tailed Pvalue were derived from the Wald test. **b**. Volcano plot highlighting differentially expressed (Pval < 0.01, FC cutoff 1) genes between sporadic- and IBD-CRCs. **c**. Box plots relative to differentially expressed CBC (Lgr5-derived) and RSC (PC-derived) marker genes between sporadic- and IBD-CRCs. N = 14 IBD-CRCs and N = 38 sCRCs. Boxplots display the median, lower and upper hinges correspond to the first and third quartiles. Whiskers extend from the hinges to max/min values, no further than 1.5*inter-quartile range. P values denote results of the t test. **d**. Bar plot relative to the gene set enrichment analysis between sCRC and IBD-CRC (abs NES > 0.5, Pval < 0.05). **e**. Box plots relative to stem cell index values across the colitis-like (N = 798), sporadic-like (N = 940), and intermediate (N = 1494) group of colon cancers. Boxplots display the median, lower and upper hinges correspond to the first and third quartiles. Whiskers extend from the hinges to max/min values, no further than 1.5*inter-quartile range. P value depicts result of one-way ANOVA.

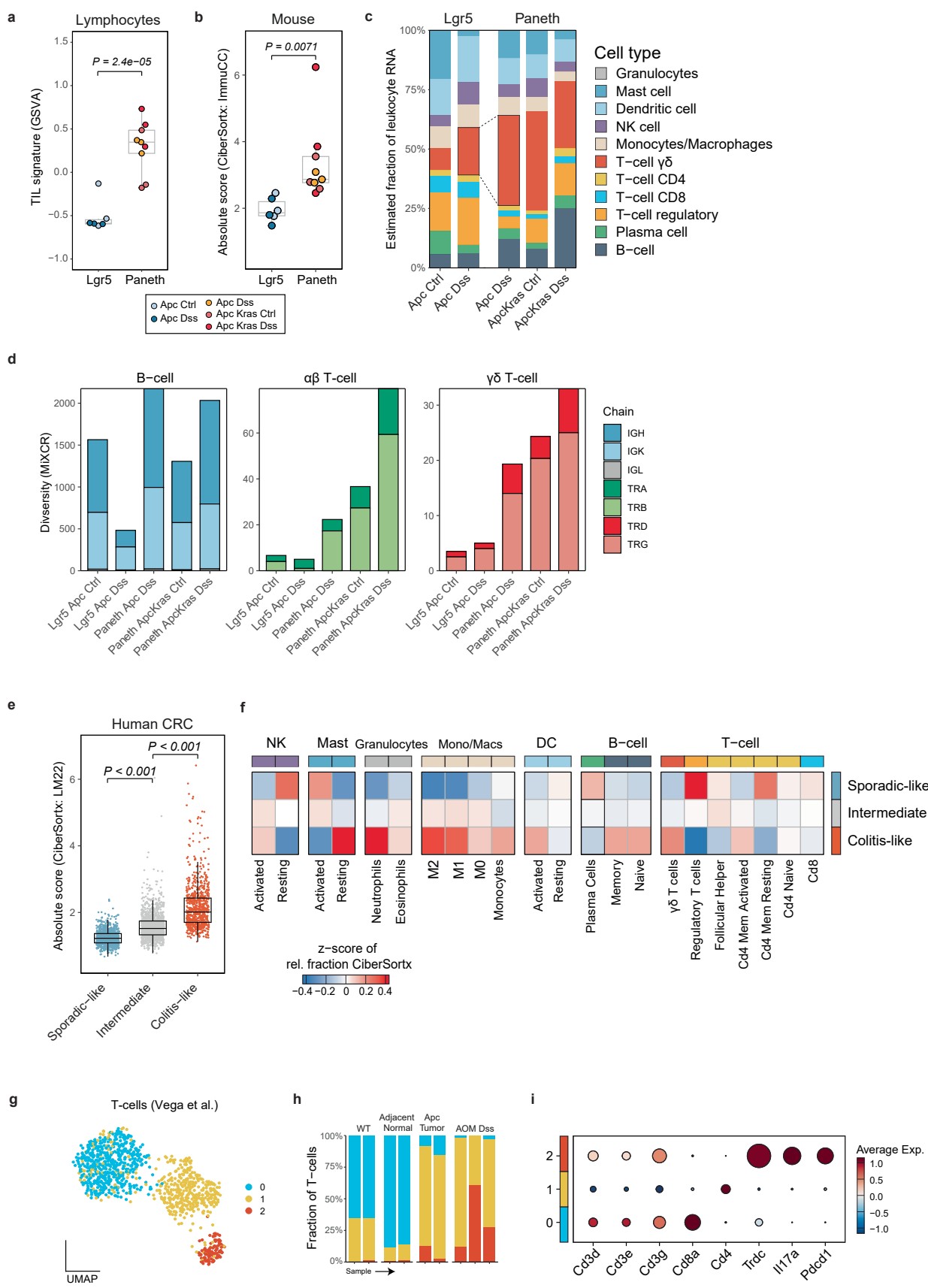

**Extended Data Fig. 8 | See next page for caption.**

**Extended Data Fig. 8 | Immune profiling of CBC- and PC-derived small intestinal adenomas. a**. Boxplot showing gene set variation score of the tumour infiltrating lymphocyte (TIL) signature[40]. N = 6 Lgr5-derived tumours and N = 9 Paneth-derived tumours. Boxplots display the median, lower and upper hinges correspond to the first and third quartiles. Whiskers extend from the hinges to max/min values, no further than 1.5*inter-quartile range. Pvalue denotes the result of two-sided t-test. **b**. CiberSortx prediction of absolute immune infiltrate based on the ImmuCC signature matrix. N = 6 Lgr5-derived tumours and N = 9 Paneth-derived tumours. Boxplots display the median, lower and upper hinges correspond to the first and third quartiles. Whiskers extend from the hinges to max/min values, no further than 1.5*inter-quartile range. Pvalue denotes the result of two-sided t-test. **c**. Stacked bar plot denoting the relative contribution of leukocyte subtypes in CBC- and PC-derived intestinal adenomas. N = 3 samples per group. **d**. Bar plots showing the number of immune receptor clonotypes, separated by chain usage, across CBC- and PC-derived tumours.

Bars denote the average of N = 3 samples. **e**. CiberSortx prediction of absolute immune infiltrate based on the LM22 signature on human CRCs, stratified according to their sporadic-like and colitis-like expression profiles. N = 798 colitis-like CRCs, N = 1494 intermediate and N = 940 sporadic-like CRCs. Boxplots display the median, lower and upper hinges correspond to the first and third quartiles. Whiskers extend from the hinges to max/min values, no further than 1.5*inter-quartile range. Pvalue denotes the result of two-sided t-test. **f**. Heat map showing relative differences in infiltrating immune cells between sporadic- and colitis-like CRCs. g. UMAP dimension reduction displaying T-cells in murine tumours from Vega et al.[33] subdivided in three distinct sub-clusters. **h**. Stacked bar plot indicating the distribution of the T-cell sub-clusters among the Lrig1-Apc and AOM/DSS-derived tumours. **i**. Dot plot showing the expression profile of discriminating markers across the T-cell sub-clusters. P values denote results of t tests.

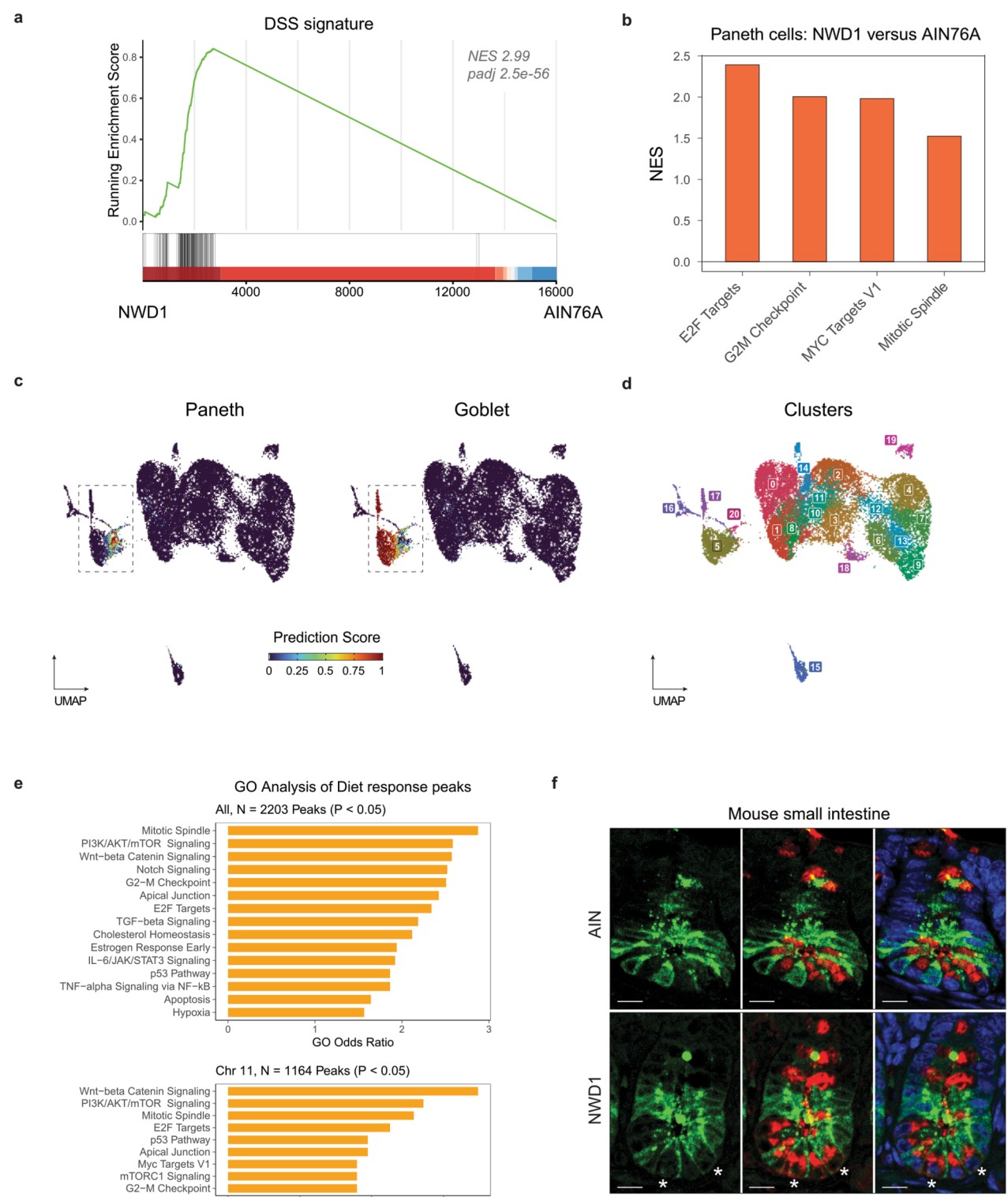

**Extended Data Fig. 9 | The NWD1 western-style diet triggers the dedifferentiation of Paneth cells. a.** GSEA plot of Paneth cells from NWD1- vs. AIN76A-fed mice. The DSS signature significantly associates with the NWD1 diet (NES 2.99, Padj 2.5e-56). P value denote Benjamini-Hochberg adjusted value of the two-sided enrichment P-value from GSEA. **b.** Bar plot showing pathways elevated (P < 0.05, NES > 0.5) in Paneth cells from NWD1- when compared to AIN76A-fed animals. **c.** Prediction scores for Paneth (left) and goblet (right) cells on UMAP embedding from the scATAC data (Choi et al.[45]). **d.** Unsupervised clustering reveals two distinct clusters (c5, c20) that include Paneth cells. **e.** Bar plot listing gene ontology (GO) results based on differential peaks from the diet responsive Paneth cells. Pvalues were obtained from the EnrichR GO analysis and denote the results of Fisher's exact test **f.** Immunofluorescence imaging of WGA and Olf4m showing intestinal crypts from mice fed with AIN/NWD1 diets. Asterisks earmark double positive cells. Scale bar indicates 10 μm.

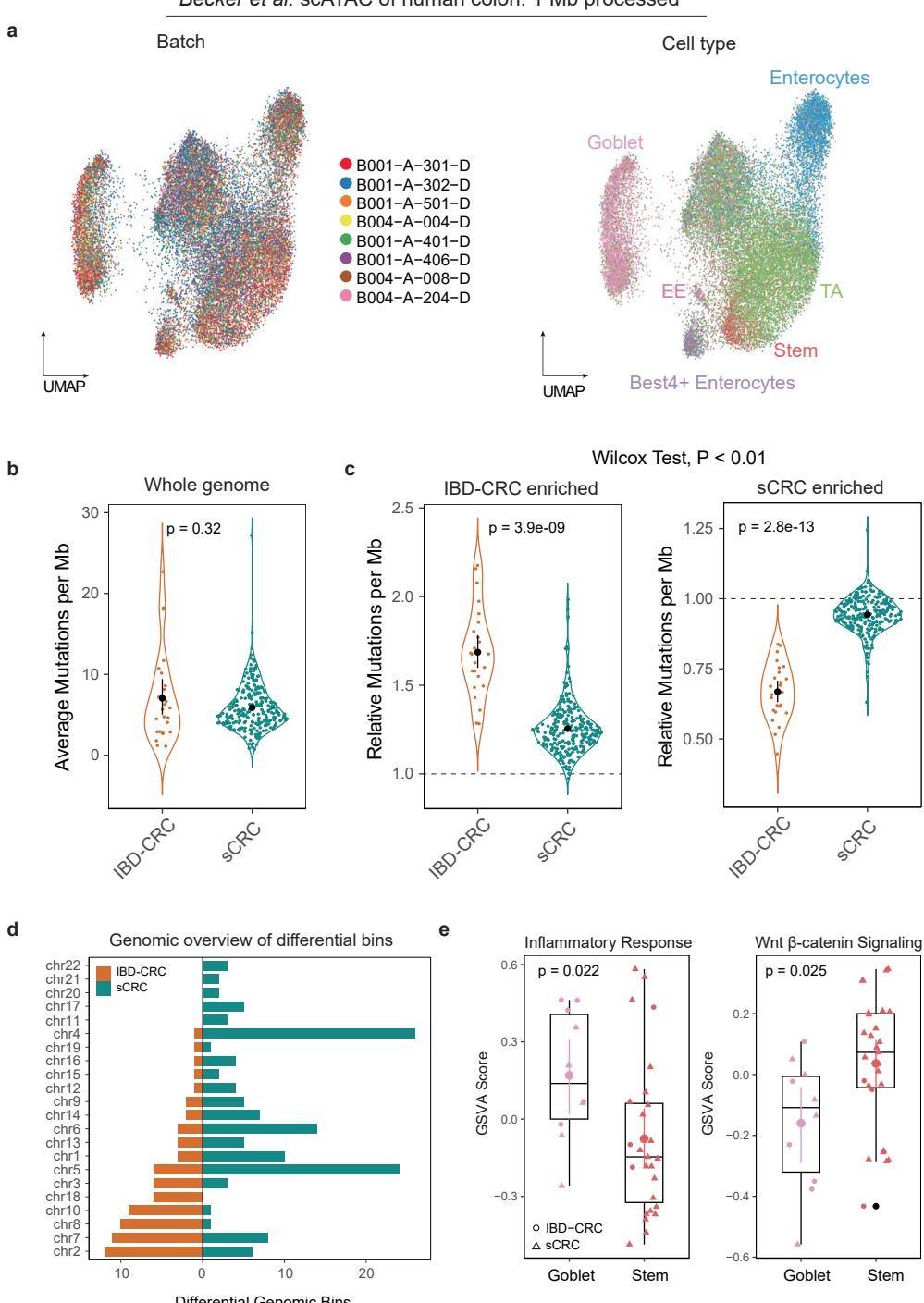

**Extended Data Fig. 10 | Epigenetic profiles of normal colonic lineages and mutational patterns of IBD-related and sporadic colon cancers. a.** UMAP plot displaying 1 Mb processed scATAC data of the human colon (Becker et al.[52]). **b**, **c**. Average number of passenger mutations (**b**) in IBD-related and sporadic colon cancers[36] N = 257 sCRCs and N = 25 IBD-CRCs. (**c**). Relative mutation frequency in IBD-enriched (N = 78) and sCRC-enriched (N = 136) bins based on two-sided Wilcox test (P < 0.01). Error bars are displayed from the mean and indicate the standard error from bootstrap with a 95% confidence interval.

**d**. Barplot showing the genomic overview of differential bins across chromosomes. e. Boxplots relative to the gene set variation analysis (GSVA) scores of tumors stratified according to their predicted cell of origin. P values denote results of t tests. N = 27 Stem-predicted and N = 10 Goblet-predicted CRCs. Boxplots display the median, lower and upper hinges correspond to the first and third quartiles. Whiskers extend from the hinges to max/min values, no further than 1.5*inter-quartile range.

# Reporting Summary

## Statistics

For all statistical analyses, confirm that the following items are present in the figure legend, table legend, main text, or Methods section.

| n/a | Confirmed | |
|---|---|---|
| ☐ | ☒ | The exact sample size (*n*) for each experimental group/condition, given as a discrete number and unit of measurement |
| ☐ | ☒ | A statement on whether measurements were taken from distinct samples or whether the same sample was measured repeatedly |
| ☐ | ☒ | The statistical test(s) used AND whether they are one- or two-sided<br>*Only common tests should be described solely by name; describe more complex techniques in the Methods section.* |
| ☐ | ☒ | A description of all covariates tested |
| ☐ | ☒ | A description of any assumptions or corrections, such as tests of normality and adjustment for multiple comparisons |
| ☐ | ☒ | A full description of the statistical parameters including central tendency (e.g. means) or other basic estimates (e.g. regression coefficient) AND variation (e.g. standard deviation) or associated estimates of uncertainty (e.g. confidence intervals) |
| ☐ | ☒ | For null hypothesis testing, the test statistic (e.g. *F*, *t*, *r*) with confidence intervals, effect sizes, degrees of freedom and *P* value noted<br>*Give P values as exact values whenever suitable.* |
| ☒ | ☐ | For Bayesian analysis, information on the choice of priors and Markov chain Monte Carlo settings |
| ☒ | ☐ | For hierarchical and complex designs, identification of the appropriate level for tests and full reporting of outcomes |
| ☐ | ☒ | Estimates of effect sizes (e.g. Cohen's *d*, Pearson's *r*), indicating how they were calculated |

*Our web collection on statistics for biologists contains articles on many of the points above.*

## Software and code

Policy information about availability of computer code

| | |
|---|---|
| Data collection | Histological analysis: Nanozoomer (Hamamatsu), NDP Viewer (v2, Hamamatsu).<br>Fluorescent images: Confocal LSM 700 (Zeiss), Confocal Stellaris 5 (Leica).<br>Real Time PCRs: 7500 Real Time System (Applied Biosystems).<br>FACS: FACSAria III (BD Biosciences)<br>scRNAseq: Chromium Controller (10X Genomics).<br>sequencing: HiSeq 2500 (Illumina), NovaSeq 6000 (Illumina), DNBSEQ-G400 (MGI). |
| Data analysis | Statistical analysis was performed in R (v4.2.1) running under Red Hat Enterprise Linux 8.6.<br>For analysis of histology and fluorescent pictures, the following software packages were used: ImageJ (v 2.0.0), QuPath (v 0.4.0), NDPViewer (v 2.9.25)<br>For analysis of FACS data, the following software packages were used: FACSDiva (v.8.0.1, BD Biosciences), FlowJo (V10).<br>Analysis of transcriptomic data was based on publicly available software packages as specified in Methods. Scripts are available upon request.<br>For bulk RNA seq analysis, the following software packages were used: SOAPnuke pipeline (v1), STAR (v 2.7.9a), sambamba (v 0.8.0), subread (v 2.0.3), DESeq2 (v 1.34.0), biomaRt (v 2.52.0), GSVA (v 1.44.5), fgsea (v 1.22.0), ggplot2 (3.4.0), dplyr (v 1.0.10), ComplexHeatmap (v.2.12.1), EnhancedVolcano (v 1.14.0), ISCindex (v 0.0.0.9), survival (v 3.3-1), survminer (v 0.4.9), CiberSortx, MiXCR (v4.6.0), immunarch (v0.9.0).<br>For scRNA seq analysis, the following software packages were used: cellranger (v 7.0.0), Seurat (v 4.1.1), DoubletFinder (v 2.0.3), biomaRt (v 2.52.0), GSVA (v 1.44.5), fgsea (v 1.22.0), ggplot2 (3.4.0), dplyr (v 1.0.10), ComplexHeatmap (v.2.12.1), CytoTRACE (v 0.3.3).<br>For scATACseq analysis, the following packages were used: Seurat (v 4.1.1), Signac (v 1.9.0), GenomicRanges (v 1.50.2), RIdeogram (v 0.2.2). |

For manuscripts utilizing custom algorithms or software that are central to the research but not yet described in published literature, software must be made available to editors and reviewers. We strongly encourage code deposition in a community repository (e.g. GitHub). See the Nature Portfolio guidelines for submitting code & software for further information.

# Data

Policy information about availability of data

All manuscripts must include a data availability statement. This statement should provide the following information, where applicable:
- Accession codes, unique identifiers, or web links for publicly available datasets
- A description of any restrictions on data availability
- For clinical datasets or third party data, please ensure that the statement adheres to our policy

All data relevant to this study are made available. Transcriptomic sequencing data has been deposited in the Gene Expression Omnibus (GEO), and is available using the following identifiers: GSE221819 (bulk RNA sequencing of murine tumors); GSE221820 (single cell RNA sequencing of genetically targeted Paneth cells); GSE221818 (bulk RNA sequencing of sorted Paneth cells treated with control and western-style diet). Additional data sets referenced in this study are publicly available in the Synapse and GEO repositories.
Previously published scRNAseq studies were employed relative to the mouse small intestine in homeostasis (Haber et al., GSE92332), upon irradiation (Ayyaz et al., GSE117783), Lgr5 ablation (Singh et al., GSE183299), and upon feeding with western-style diet (Choi et al., scRNAseq: GSE188577, scATACseq: GSE228006). Mouse colonic scRNAseq data from AOM/DSS tumors was obtained from Vega et al. (GSE134255) and bulk RNAseq was from Chen et al. (GSE178145). Human colorectal cancer Bulk RNAseq and scRNAseq studies were retrieved from Guinney et al. (syn2623706), Rajamäki et al. and Lee et al. (KUL cohort: GSE144735; SMC cohort: GSE132465), respectively. Human scRNAseq data from Ulcerative Colitis patients (Smillie et al.) was downloaded from the Broad DUOS platform after a DTA agreement. Single cell ATAC data from human colon was obtained from Becker et al. (GSE201349).

# Human research participants

Policy information about studies involving human research participants and Sex and Gender in Research.

| | |
|---|---|
| Reporting on sex and gender | NA |
| Population characteristics | NA |
| Recruitment | NA |
| Ethics oversight | The study followed the guidelines of the European Network of Research Ethics Committees, in line with European, national, and local regulations. As per national protocols, informed consent was not required for the immune-histological analysis of residual tissue material. Approval for the use of material from IBD patients was obtained from the medical ethical committee of the Erasmus MC under license MEC-2009-041. |

Note that full information on the approval of the study protocol must also be provided in the manuscript.

# Field-specific reporting

Please select the one below that is the best fit for your research. If you are not sure, read the appropriate sections before making your selection.

☒ Life sciences   ☐ Behavioural & social sciences   ☐ Ecological, evolutionary & environmental sciences

For a reference copy of the document with all sections, see nature.com/documents/nr-reporting-summary-flat.pdf

# Life sciences study design

All studies must disclose on these points even when the disclosure is negative.

| | |
|---|---|
| Sample size | For the generation of novel mouse models, no statistical test was used to determine the sample size upfront. As these models were evaluated for the first time, there was no data available upfront to predict the needed sample size in a reliably manner. Moreover, the variation in litter size and breeding efficiency of complex genotypes limits the control of experimental group numbers. For in vivo tumorigenesis experiments, at least 10 animals were screened per genotype, except for Lyz1/Apc/Kras/P53 animals due to difficulties in breeding animals with this genotype. RNAseq analysis was done on N = 3 tumors per group, which we considered as statistical minimum given a large effect size. Single cell RNAseq was performed on at least N = 3 animals per genotype, except for Lyz1/Kras and Lyz1/Apc/Kras/P53, that were analyzed for N = 2 animals. Diet scRNAseq data was obtained from N = 3 mice per group, and scATACseq data was obtained from N = 2 mice per group. In vitro organoid reconstitution assays were performed in duplicate on N = 4 mice per group. |
| Data exclusions | One mouse has been excluded from the scRNAseq analysis, as the number of sorted Yfp cells was too low (below 50 by FACS counts, possibly due to poor Tamoxifen administration) to draw significant conclusions. Occasionally, mice reached their pre-established humane endpoint (e.g. 20% weight loss) after administration of DSS, which led to exclusion of those mice from the study.<br>No human samples have been excluded from the analysis. An exception is the analysis regarding consensus molecular subtypes. Tumors that were classified as 'unknown' (due to non-significant CMS classification) were filtered from the analysis. |
| Replication | All experiments have been repeated at least N = 3 times, except the scRNAseq analysis of mice with Lyz1-AKP (N = 2) and Lyz1-K genotype (N = 2). Additional validation of results obtained from these samples is provided by corroborative methods, such as flow cytometry analysis, histological analysis and in situ hybridizations. Attempts at replication were successful for the experiments. |

| | |
|---|---|
| Randomization | Mice were randomly assigned to experimental groups after validation of their genotype, and controlling for age and sex. |
| Blinding | Whenever possible, experimental labels were blinded to the collectors of the data, and labels were added post data collection during the analyses. |

# Reporting for specific materials, systems and methods

We require information from authors about some types of materials, experimental systems and methods used in many studies. Here, indicate whether each material, system or method listed is relevant to your study. If you are not sure if a list item applies to your research, read the appropriate section before selecting a response.

## Materials & experimental systems

| n/a | Involved in the study |
|---|---|
| ☐ | ☒ Antibodies |
| ☒ | ☐ Eukaryotic cell lines |
| ☒ | ☐ Palaeontology and archaeology |
| ☐ | ☒ Animals and other organisms |
| ☒ | ☐ Clinical data |
| ☒ | ☐ Dual use research of concern |

## Methods

| n/a | Involved in the study |
|---|---|
| ☒ | ☐ ChIP-seq |
| ☐ | ☒ Flow cytometry |
| ☒ | ☐ MRI-based neuroimaging |

## Antibodies

| | |
|---|---|
| Antibodies used | Primary (mouse):<br>β-catenin (#610154, BD Biosciences)<br>γ-H2AX (#9718, Cell Signaling)<br>Gfp (#A-11122, ThermoFisher)<br>Olfm4 (#D6Y5A, Cell Signaling)<br>Lyz1 (#A0099, Dako)<br>Dclk1 (#ab37994, Abcam)<br>Muc2 (#sc-15334, Santa Cruz)<br>Chga (#NB120-15160, Novus Bio)<br>Ecad (#610182, BD Biosciences)<br>Mki67 (#NB500, Novus Biologicals)<br>Cd3 (#ab5690, Abcam)<br>F4/80 (#D2S9R, Cell Signaling)<br>αSMA (#ab5690, Abcam)<br><br>Primary (human)<br>KI67 (MIB-1 antibody, #790-4286; Ventana)<br>MUC2 (CCP58, #M7313, Dako)<br>BEST4 (HPA058564, Sigma-Aldrich)<br><br>Fluorescent<br>Goat anti-Rabbit IgG (H+L) secondary Antibody Alexa Fluor® 488 conjugate (#A32731, Invitrogen)<br>anti-rabbit Alexa 568 (#A11011, Invitrogen)<br>anti-mouse Alexa 633 (#A21052, Invitrogen)<br>phalloidin Alexa 647 (#A22287, Invitrogen)<br>SiR-actin (#SC001, Spirochrome)<br>Wheat Germ Agglutinin (#W32466, ThermoFisher Scientific)<br><br>FACS<br>CD31-BV421 (#563356, BD Biosciences)<br>CD45-BV421 (#563890, BD Biosciences)<br>TER119-BV421 (#563998, BD Biosciences)<br>CD24-APC (#1109070, Sony Biotechnology)<br>cKit-PE (#105808, Biolegend)<br>cKit-BB515 (#564481, BD Horizon)<br><br>FACS (Hashing)<br>Hashtag 1 Antibody (#155831, Biolegend)<br>Hashtag 2 Antibody (#155833, Biolegend)<br>Hashtag 4 Antibody (#155837, Biolegend)<br>Hashtag 7 Antibody (#155843, Biolegend)<br>Hashtag 10 Antibody (#155849, Biolegend) |
| Validation | All antibodies used in this study has been validated by the providers by western blot analysis and stainings on cell-lines/tissues known to produce the particular protein. Antibodies were titrated to optimize the stainings on sections of the mouse small intestine. |

# Animals and other research organisms

Policy information about studies involving animals; ARRIVE guidelines recommended for reporting animal research, and Sex and Gender in Research

| Laboratory animals | For all experiments, C57BL/6 mice were randomly assigned to experimental groups after matching for gender, age of 8-12 weeks, and genotype. All protocols involving animals were approved by the Dutch Animal Experimental Committee and in accordance with the Code of Practice for Animal Experiments in Cancer Research established by the Netherlands Inspectorate for Health Protections, Commodities and Veterinary Public Health. Animals were bred and maintained in the Erasmus MC animal facility (EDC) under conventional specific pathogen-free conditions. Dark/light cycles were maintained between 7PM-7AM (dark) and 7AM-7PM (light); humidity was kept around 50% (with min 45%, max 65%) and temperature between 20-24°C. The following strains were used and bred with different combinations: Lgr5CreERT2-EGFP (#008875, Jackson Lab) pLysCreERT2 (kind gift from H. Clevers Lab) R26LSL-tdTomato (#007908, Jackson Lab) R26LSL-YFP (#006148, Jackson Lab) Apc15lox (#029275, Jackson Lab) KrasLSL-G12D (#008179, Jackson Lab) Tp53flox (#008462, Jackson Lab) Lgr5DTR-EGFP (MGI:5294798, kind gift from F. de Sauvage Lab) c-KitCreERT2 (MGI:5543260, kind gift from D. Saur Lab) |
|---|---|
| Wild animals | The study did not involve wild animals |
| Reporting on sex | All experiments have been performed with male and female animals. Wherever possible, mice were assigned to experimental groups to ensure equal distribution of sexes. Characteristics of the mice including sex will be added to the Figure source data. |
| Field-collected samples | The study did not involve samples collected from the field |
| Ethics oversight | All protocols involving animals were approved by the Dutch Animal Experimental Committee and in accordance with the Code of Practice for Animal Experiments in Cancer Research established by the Netherlands Inspectorate for Health Protections, Commodities and Veterinary Public Health. |

Note that full information on the approval of the study protocol must also be provided in the manuscript.

# Flow Cytometry

## Plots

Confirm that:

☒ The axis labels state the marker and fluorochrome used (e.g. CD4-FITC).

☒ The axis scales are clearly visible. Include numbers along axes only for bottom left plot of group (a 'group' is an analysis of identical markers).

☒ All plots are contour plots with outliers or pseudocolor plots.

☒ A numerical value for number of cells or percentage (with statistics) is provided.

## Methodology

| Sample preparation | Crypts were purified from the small intestine, and enzymatically dissociated with TryplE to obtain a single cell suspension. Cells were stained with fluorescent antibodies (Cd24, cKit, Cd31, Cd45, Ter119) and with Hashing antibodies. Paneth cells were enriched by Fluorescence-activated cell sorting (FACS) according to the SSChiCD24hicKithi gate or by the presence of Yfp/td-Tomato signal |
|---|---|
| Instrument | FACSAriaIII (BD Biosciences) |
| Software | Raw data has been processed with FACSDiva™, and the gating strategy was visualized with FlowJo™ v10. |
| Cell population abundance | Crypt-purified single cell preparations consisted of 2-6% Paneth cells, defined by SSC/Cd24/cKit (see Suppl. Fig. 4b). The purity of post-sorted cells was evaluated by re-analysis with FACS (% of sorted cells mapping in same gate). This purity assessment was in the range of 97-99% (for detailed protocol, see Schewe et al. 2017, JOVE). Viability post sorting, reanalyzed by FACS using DAPI at different time points post sorting till 2 hours (on ice), was > 80-95% for Paneth cells. |
| Gating strategy | The gating strategy is visualized in Suppl. Fig. 3. In brief, live epithelial cells were obtained by filtering out debris, dead cells and doublets (top panels), Subsequently, Paneth cells were identified based on high SSC-A and high Cd24. This gate was further purified based on high cKit levels. Traced cells were sorted based on high levels of either Yfp or td-Tomato. |

☒ Tick this box to confirm that a figure exemplifying the gating strategy is provided in the Supplementary Information.

