## [Peer Review File · Nature Genetics]

Peer Review Information

Manuscript Title: The non-stem origin of colon cancer in the context of inflammation

Corresponding author name(s): Professor Riccardo Fodde

Editorial Notes:

Transferred manuscripts This document only contains reviewer comments, rebuttal and decision letters for versions considered at Nature Genetics.

Reviewer Comments & Decisions:

Decision Letter, initial version:

23rd Jan 2024

Dear Professor Fodde,

First, please accept my apologies for the delay in returning this decision to you. Thank you for your patience.

Your Article, "The origin of intestinal cancer in the context of inflammation" has now been seen by the 3 referees who originally reviewed the paper for Nature (the numbering of reviewers remains unchanged). You will see from their comments below that while they find your work of interest, some important points are raised. We are interested in the possibility of publishing your study in Nature Genetics, but would like to consider your response to these concerns in the form of a revised manuscript before we make a final decision on publication.

We therefore invite you to revise your manuscript taking into account all reviewer and editor comments. Please highlight all changes in the manuscript text file. At this stage we will need you to upload a copy of the manuscript in MS Word .docx or similar editable format.

*2) If you have not done so already please begin to revise your manuscript so that it conforms to our Article format instructions, available here.
Refer also to any guidelines provided in this letter.

Please be aware of our guidelines on digital image standards.

[redacted]

We hope to receive your revised manuscript within four to eight weeks. If you cannot send it within this time, please let us know.

Sincerely,

Safia Danovi
Editor
Nature Genetics

Reviewers' Comments:

Reviewer #1:

Remarks to the Author:

As I mentioned in my previous review of the Verhagen et al paper, the study is well-executed, and the results are intriguing. However, I raised several concerns about the strength and scope of the author's conclusions regarding the role of Paneth cells as the unique cell of origin of CRC under inflammatory conditions. The authors have provided extensive arguments in their rebuttal letter to discuss my criticisms, but a majority of them have not been experimentally addressed. Instead, they have introduced changes in the revised version, including altering the title and re-writing parts of the manuscript, to convey my concerns. I also acknowledge the new bioinformatic inference analyses of the cell of origin of CRC from patient genomic datasets, which further supports the author's observations. The data showing proliferation deep crypt secretory cells of the colon in Western diet conditions also reinforces the data. I'd like to see a further revised version that includes a more thorough and explicit discussion of the study's limitations, encompassing the various issues tackled in the rebuttal letter:

-“Paneth cell has always been intended as a model of a fully committed and post-mitotic lineage capable of de-differentiation and tumorigenesis upon inflammation.” Please clearly state that Paneth cells may likely be just one of the many contributors to intestinal cancer under inflammation and that other differentiated cell types, such as goblet cells or enterocytes may give rise to RSCs through plasticity and therefore originate cancers upon accumulation of driver mutations.”

- Acknowledgment of the possibility that Lgr5+ ISCs can also contribute to CRC in an inflammatory setting should be made, given that they are not entirely eliminated in these conditions (and also as reflected by the computational predictions included in this revised version - Figure 5d).

Reviewer #2:

Remarks to the Author:

The authors have addressed my concerns and I recommend the manuscript be accepted.

Reviewer #3:

Remarks to the Author:

With this revised version, the authors have addressed many of my comments and improved the quality of the manuscript. However, there remains 3 issues that need to be addressed prior to publication in Nature Genetics. The concerns are outlined below.

Point 1:

In the rebuttal, the authors suggest that endogenous Lyz1 mRNA expression is only present in PCs and thus without evidence to the contrary there is no justification in performing the “complex” in vivo

experiments that were proposed. Firstly, the original experiment asked for, i.e. induction of Apc deletion in homeostatic Paneth cells 20 days prior to DSS treatment instead of 1 day after DSS treatment, is by no means complex. This could have been done relatively quickly. Secondly the point raised about not observing endogenous Lyz1 expression in any other cell type other than PCs is well taken but does not rule out the possibility of ectopic induction of the reporter Cre line in regenerative cells. There are many examples of Cre lines not entirely recapitulating the behaviour of their endogenous genes. Inducing Apc deletion in PCs 20 days prior would provide sufficient time for systemic tamoxifen to be removed and therefore entirely avoid the possibility of ectopic deletion in cells other than PCs post DSS. To further avoid these potential effects of systemic tam, the authors could also administer pre-DSS low dosages (<1mg per 20g) of tamoxifen. If the authors observe tumor initiation under these conditions this would strongly support their conclusions. The absence of tumors under these conditions and confirmation that Apc was deleted in isolated Paneth cells would raise doubts as to the validity of their model.

Finally, the fact that Apc and Kras mutants develop tumors from homeostatic PCs does support their overall claims. But it remains unclear whether simultaneous activation of multiple oncogenic pathways in a differentiated cell type models tumorigenesis in humans. For this reason, a clear demonstration that Apc deletion alone coupled to injury or colitis is sufficient to drive tumorigenesis from PCs is important.

Point 2:

The scRNAseq data in Fig 2c appears to indicate that following DSS treatment, wt PCs undergo reprogramming events but these do not include revival stem cell activation since cluster 4 is not induced upon DSS alone. This would indicate that the cluster 4 signature is secondary to APC/Kras mutations; whereas the regenerative capacity of Paneth cells (in the absence of oncogenic signals) is independent of a transition to a revival stem cell state. However elsewhere in the manuscript, re-analysis of the single cell data following ablation of Lgr5+ cells (Singh et al. paper) showed that PCs induce a revival stem cell signature. So why are wild-type Paneth cells not activating a revival cell signature in the context of DSS but are in the context of Lgr5+ cell ablation? A discussion of this point could also be included in the manuscript.

On a related note, a summarizing statement in the abstract mentions the following: "By combining scRNAseq with lineage tracing to capture the conversion of PCs into bona fide tumor cells, we show that they progress through a "revival stem cell" (RSC) state characterized by high Clusterin (Clu) expression and Yap1 signaling, reminiscent of what has previously been observed upon irradiation." This may be the case, but a similar transition occurs in ISC cells as they transition to adenomas. Indeed, there are many papers showing that ISC derived tumor cells express revival cell markers. Thus the author's analysis does not reveal new insight into how Paneth cells are reprogrammed to tumor cells. What are the unique trajectories that Paneth cells take, relative to Lgr5+ISC, that lead to the generation of such completely different tumor types? Again at the very least this point should be discussed in the manuscript.

Point 3.

To confirm the inflammatory signature in PC derived tumors, the authors examine the presence of inflammatory cells (i.e. T cells, Macs, Fib) in PC tumors vs Lgr5 cell tumors. Stainings were performed in Supplementary Figure 5 but the differences they reveal are not clear at all. In particular the CD3 staining does not seem to change at all between the various tumor types. This would suggest that the inflammatory signature may indicate qualitative differences in the type of immune or stromal cells that are recruited to PC vs Lgr5 tumors given that generic markers of stromal, macrophages and T cells

apparently show no difference. The heatmap in Fig 3h shows certain markers of immune cell or stromal cells are upregulated in PC tumors/IBD. Do these signatures reveal functional differences or specific subpopulations of immune and stromal cells of interest ? For instance, M1 vs M2 macrophages, CD8 vs CD4 T cells, myofibroblasts vs trophocytes etc. etc. This needs to be investigated further and/or these points should be raised in a proper discussion.

Author Rebuttal to Initial comments

Response to referees relative to ms. NG-A64104-T Fodde

Reviewer #1

“Paneth cell has always been intended as a model of a fully committed and post-mitotic lineage capable of de-differentiation and tumorigenesis upon inflammation.” Please clearly state that Paneth cells may likely be just one of the many contributors to intestinal cancer under inflammation and that other differentiated cell types, such as goblet cells or enterocytes may give rise to RSCs through plasticity and therefore originate cancers upon accumulation of driver mutations.”

Re: we have never wrote or even implied that Paneth cells were the only contributors to intestinal cancer upon inflammation. As stated in the ms, we employed the Paneth lineage as a model of a fully-differentiated and post-mitotic cell type capable of re-entering the cell cycle, dedifferentiating, and tumor forming upon tissue injury. Accordingly, our new data obtained from the machine-learning analysis of the whole-genome mutation spectra of patient-derived colon cancers, clearly show that different lineages are likely to represent the cell-of-origin of bowel malignancies in the majority of IBD-related cases and in up to 40% of the sporadic burden. In compliance with the reviewer’s request, these concepts were clarified and reiterated in the Discussion of the revised manuscript.

- Acknowledgment of the possibility that Lgr5+ ISCs can also contribute to CRC in an inflammatory setting should be made, given that they are not entirely eliminated in these conditions (and also as reflected by the computational predictions included in this revised version - Figure 5d).

Re: Both in the Results and Discussion sections of the revised manuscript, we have now stated that, upon DSS-driven inflammation, Lgr5⁺ ISC ablation is not complete and as such ISCs may also contribute to tumor formation and to increased intra-tumor heterogeneity, a known predictor of resistance to conventional therapy and of poor overall survival among colon cancer patients.

Reviewer #3:

Point 1:

In the rebuttal, the authors suggest that endogenous Lyz1 mRNA expression is only present in PCs and thus without evidence to the contrary there is no justification in performing the "complex" in vivo experiments that were proposed. Firstly, the original experiment asked for, i.e. induction of Apc deletion in homeostatic Paneth cells 20 days prior to DSS treatment instead of 1 day after DSS treatment, is by no means complex. This could have been done relatively quickly. Secondly the point raised about not observing endogenous Lyz1 expression in any other cell type other than PCs is well taken but does not rule out the possibility of ectopic induction of the reporter Cre line in regenerative cells. There are many examples of Cre lines not entirely recapitulating the behaviour of their endogenous genes. Inducing Apc deletion in PCs 20 days prior would provide sufficient time for systemic tamoxifen to be removed and therefore entirely avoid the possibility of ectopic deletion in cells other than PCs post DSS. To further avoid these potential effects of systemic tam, the authors could also administer pre-DSS low dosages (<1mg per 20g) of tamoxifen. If the authors observe tumor initiation under these conditions this would strongly support their conclusions. The absence of tumors under these conditions and confirmation that Apc was deleted in isolated Paneth cells would raise doubts as to the validity of their model.

Finally, the fact that Apc and Kras mutants develop tumors from homeostatic PCs does support their overall claims. But it remains unclear whether simultaneous activation of multiple oncogenic pathways in a differentiated cell type models tumorigenesis in humans. For this reason, a clear demonstration that Apc deletion alone coupled to injury or colitis is sufficient to drive tumorigenesis from PCs is important.

Re: Relative to the complexity of the proposed experiment, since the obligatory readout is tumor onset and multiplicity, its completion will take 8-9 months. Apart from this, as also agreed by the reviewer, the very fact that combined Apc and Kras mutations targeted in Paneth cells do result in multiple tumors even in the absence of DSS, is in full support of our claims and makes the hypothetical scenario where illegitimate Cre-driven recombination occur as the result of the DSS-driven inflammation, extremely unlikely. As oncogenic and PC-specific activation of Kras alone, with or without DSS, does not result in any tumors, whereas Apc does, makes this hypothetical scenario even more unlikely and simply does not justify a delay of at least 8 months.

Last but not least, as previously explained to the editor, at present we do not have the necessary license to perform these experiments.

Point 2:

The scRNAseq data in Fig 2c appears to indicate that following DSS treatment, wt PCs

undergo reprogramming events but these do not include revival stem cell activation since cluster 4 is not induced upon DSS alone. This would indicate that the cluster 4 signature is secondary to APC/Kras mutations; whereas the regenerative capacity of Paneth cells (in the absence of oncogenic signals) is independent of a transition to a revival stem cell state. However elsewhere in the manuscript, re-analysis of the single cell data following ablation of Lgr5+ cells (Singh et al. paper) showed that PCs induce a revival stem cell signature. So why are wild-type Paneth cells not activating a revival cell signature in the context of DSS but are in the context of Lgr5+ cell ablation? A discussion of this point could also be included in the manuscript.

Re: We thank the reviewer for his/her comment on this specific issue. Indeed, RSC activation was mainly observed upon the combined Apc/Kras mutations. However, as shown in Figure 2e, the use of a RSC signature showed increased expression also in the other PC clusters. The differences between our results and those in the Singh et al. and Ayaz et al. studies are likely to be caused by differences in the intensity of the inflammatory insult (e.g. % of DSS administered in the drinking water; length of the treatment cycles, etc.) and by the degree of ISC ablation (close to 100% upon diphtheria toxin, and only partial by DSS). These admittedly speculative considerations have now been included in the revised version of the manuscript.

On a related note, a summarizing statement in the abstract mentions the following: "By combining scRNAseq with lineage tracing to capture the conversion of PCs into bona fide tumor cells, we show that they progress through a "revival stem cell" (RSC) state characterized by high Clusterin (Clu) expression and Yap1 signaling, reminiscent of what has previously been observed upon irradiation." This may be the case, but a similar transition occurs in ISC

cells as they transition to adenomas. Indeed, there are many papers showing that ISC derived tumor cells express revival cell markers. Thus the author's analysis does not reveal new insight into how Paneth cells are reprogrammed to tumor cells. What are the unique trajectories that Paneth cells take, relative to Lgr5+ISC, that lead to the generation of such completely different tumor types? Again at the very least this point should be discussed in the manuscript.

Re: We have now included this issue in the Discussion of the revised manuscript. Although it has indeed been shown that ISCs may progress through a RSC state, this is unlikely to be a frequent event and the statement according to which our "analysis does not reveal new insight into how Paneth cells are reprogrammed to tumor cells" is uncalled for, let alone the fact that we ourselves stated "reminiscent of what has previously been observed upon irradiation". We never claimed to be the first to report on the RSC transition, but have simply showed for the first time that Paneth cells do undergo this transition during the tumor

forming process. The reviewer may be overlooking the simple fact that we are establishing a connection between the transient RSC state and the process through which a fully committed lineage, such as Paneth cells, becomes a cell-of-origin of cancer upon inflammatory insults.

Point 3.

To confirm the inflammatory signature in PC derived tumors, the authors examine the presence of inflammatory cells (i.e. T cells, Macs, Fib) in PC tumors vs Lgr5 cell tumors. Stainings were performed in Supplementary Figure 5 but the differences they reveal are not clear at all. In particular the CD3 staining does not seem to change at all between the various tumor types. This would suggest that the inflammatory signature may indicate qualitative differences in the type of immune or stromal cells that are recruited to PC vs Lgr5 tumors given that generic markers of stromal, macrophages and T cells apparently show no difference. The heatmap in Fig 3h shows certain markers of immune cell or stromal cells are upregulated in PC tumors/IBD. Do these signatures reveal functional differences or specific subpopulations of immune and stromal cells of interest ? For instance, M1 vs M2 macrophages, CD8 vs CD4 T cells, myofibroblasts vs trophocytes etc. etc. This needs to be investigated further and/or these points should be raised in a proper discussion. **Re:**

In compliance with the reviewer's request, we have now included a more thorough *in silico* characterization of the immune profile of the TME of the mouse tumors in comparison with sporadic colon cancers. These data, included as Supplementary Figure 9, clearly show a prominent increase of $\gamma\delta^+$ T cells both in the murine PC-derived tumors and in the colitis-like patient-derived colon cancers. Other cell types (granulocytes, macrophages, etc.) seem to be upregulated in both groups which strengthen the relevance of our study for our understanding of colon cancers arisen in the context of inflammation. We further validated these observations by analyzing the scRNAseq data obtained from mouse colonic tumors induced by AOM/DSS (Vega et al.; ref #36). A subpopulation of CD8⁻/CD4⁻ and PD1⁺/IL17⁺ T cells earmarks these tumors which likely represents the counterpart of $\gamma\delta^+$ T cells observed in colitis-like colon cancers. These new data were also included in Suppl. Figure 9f-h of the revised ms.

We are grateful to the reviewer for his/her suggestion.

Decision Letter, first revision:

5th Mar 2024

Dear Dr Fodde,

Thank you for submitting your revised manuscript "The origin of intestinal cancer in the context of inflammation" (NG-A64104R). It has now been seen by the original referees and their comments are below. The reviewers find that the paper has improved in revision, and therefore we'll be happy in principle to publish it in Nature Genetics, pending minor revisions to satisfy our editorial and formatting guidelines.

Sincerely,

Safia Danovi, PhD
Senior Editor, Nature Genetics
ORCID: 0009-0007-7822-5479

Reviewer #3 (Remarks to the Author):

The authors have again improved their manuscript with this latest round of revisions. It is unfortunate that they were unable to address, for various reasons, my first point on the specificity of their Cre deleter line. This remains an unresolved issue. However considering the overall contributions of the manuscript, which are substantial, I believe the manuscript is now suitable for publication in Nature Genetics.

Final Decision Letter:

15th May 2024

Dear Dr Fodde,

I am delighted to say that your manuscript "Non-stem cell lineages as an alternative origin of intestinal tumorigenesis in the context of inflammation" has been accepted for publication in an upcoming issue of Nature Genetics.

Your paper will be published online after we receive your corrections and will appear in print in the next available issue. You can find out your date of online publication by contacting the Nature Press Office (press@nature.com) after sending your e-proof corrections.

Please note that *Nature Genetics* is a Transformative Journal (TJ). Authors may publish their research with us through the traditional subscription access route or make their paper immediately open access through payment of an article-processing charge (APC). Authors will not be required to make a final decision about access to their article until it has been accepted. Find out more about Transformative Journals

Authors may need to take specific actions to achieve compliance with funder and institutional open access mandates. If your research is supported by a funder that requires immediate open access (e.g. according to Plan S principles) then you should select the gold OA route, and we will direct you to the compliant route where possible. For authors selecting the subscription publication route, the journal's standard licensing terms will need to be accepted, including [a href="https://www.nature.com/nature-portfolio/editorial-policies/self-archiving-and-license-to-publish](https://www.nature.com/nature-portfolio/editorial-policies/self-archiving-and-license-to-publish). Those licensing terms will supersede any other terms that the author or any third party may assert apply to any version of the manuscript.

If you have not already done so, we invite you to upload the step-by-step protocols used in this manuscript to the Protocols Exchange, part of our on-line web resource, natureprotocols.com. If you complete the upload by the time you receive your manuscript proofs, we can insert links in your article that lead directly to the protocol details. Your protocol will be made freely available upon publication of your paper. By participating in natureprotocols.com, you are enabling researchers to more readily reproduce or adapt the methodology you use. [Natureprotocols.com](http://natureprotocols.com) is fully searchable, providing your protocols and paper with increased utility and visibility. Please submit your protocol to <https://protocolexchange.researchsquare.com/>. After entering your nature.com username and password you will need to enter your manuscript number (NG-A64104R1). Further information can be found at <https://www.nature.com/nature-portfolio/editorial-policies/reporting-standards#protocols>

Sincerely,

Safia Danovi, PhD
Senior Editor, Nature Genetics
ORCID: 0009-0007-7822-5479